# ADAPTING COMMUNICATING MLLMS ON THE FLY IN REFERRING EXPRESSION TASKS

## ABSTRACT

Multimodal Large Language Models (MLLMs) exhibit varying comprehension levels in language and perception that complicate interacting with a diverse population of agents, similar to how miscommunication happens in humans, e.g., because intentions are not always known. In this work, we investigate whether MLLMs can adapt to the perceptual weaknesses of the communication partners in an online manner, i.e. change the way they describe their environment in a way that is understandable to their partner while communicating with them, via reinforcement learning. We experiment with two tasks: referring expression identification (REI) and referring expression segmentation (RES), where a speaker agent has to describe an object, and a listener has to identify it. To be successful, the speaker agent must discern the comprehension level of the listener and adapt accordingly, especially when the listener suffers from perceptual weaknesses such as color blindness or blurred vision. Unlike traditional offline alignment methods for LLMs, we fine-tune a Multimodal LLM (MLLM) online to adapt to other agents' conceptual understanding. Our experiments with four MLLMs on four datasets show that online adaptation is feasible in both REI and RES settings.

## 1 INTRODUCTION

Large Language Models (LLMs) and by extension Multimodal Large Language Models (MLLMs) have demonstrated remarkable capabilities across a variety of tasks (Bubeck et al., 2023; Piergiovanni et al., 2023; Alayrac et al., 2022; Team, 2024; Anil et al., 2023) When catering MLLMs with different architectures, e.g. vision backbones, language backbones, trained with different datasets etc, in our daily lives, we may notice the variability in their comprehension levels related to task-specific concepts, i.e. what resonates with some MLLMs might not be clear to others. Disparities may exist both in their natural language understanding, e.g., some might understand expert terminology while another might require descriptive explanations, and in the perceptual understanding of visual information, e.g., some might have disabilities such as blurred vision or color blindness.

In this work, we focus on enabling MLLMs to adapt to perceptual misunderstandings of their communication partners, e.g., not perceiving colors correctly and therefore not responding to color attributes presented to them. Specifically, we fine-tune the MLLM online, i.e. on-the-fly, while it is interacting with another MLLM, based on its observed behavior. We model sequential interactions between pairs of agents during a vision-language referring expression tasks which is used as an environment for both adaptation and evaluation. Given one or two images, the speaker agent needs to describe the discriminating features of a target object, while the listener agent has to identify the correct object based on this description. To enhance overall task performance, the speaker has to learn which feature of the image allows the listener agent to discriminate the target object and adapt its communication based on the visual concepts understood by the listeners. We consider a referring expression identification (REI) task, where the listener has to identify one target image from a set of two images, and a referring expression segmentation (RES) task, where the listener has to segment the target object within a single image correctly. We present both settings in Fig. 1.

We employ several open-source MLLMs, namely LLaVA-7B, LLaVA-13B (Liu et al., 2023b), Qwen (Bai et al., 2023), and PaliGemma (Beyer et al., 2024) as the speaker and listener agents where the difference in MLLM capabilities and pre-training datasets simulate significant diversity. In addition, we introduce perceptual weaknesses to some listeners by providing them with blurred

or grayscaled images to further increase listener variety. As the benchmark, we take inspiration from Corona et al. (2019), but create a more realistic setting by modeling the interactions as free-form text, adding image transformations to simulate challenging adaptation scenarios, and scaling it to MLLMs. We evaluate the REI task on CLEVR (Johnson et al., 2017), CUB (Wah et al., 2011), and ImageNet (Deng et al., 2009), while we use the RefCOCO (Kazemzadeh et al., 2014) dataset to implement the RES task. We adapt the MLLMs on the fly using PPO (Schulman et al., 2017), KTO (Ethayarajh et al., 2024), and NLPO (Ramamurthy et al., 2023) developed originally as preference learning methods for LLMs when fine-tuning the LoRA adapters (Hu et al., 2022). Contrary to the typical use case of these algorithms for preference optimization (Ouyang et al., 2022; Ahmadian et al., 2024) where a carefully curated offline dataset of human preferences is collected, we test their efficacy during online interactions which is a more realistic and noisier setting.

Our contributions are as follows: 1) We introduce a flexible framework for evaluating four MLLMs and adapting them on the fly using four RL algorithms on natural-language-based communication tasks on four datasets to test their efficacy in online adaptation to a diverse set of communication partners. 2) We provide insights into the decision-making process of MLLMs finding that concepts related to color and shape are most important for performing well on these tasks. 3) Through extensive experimental results on two different communication tasks, four MLLMs, and four datasets, we show that adaptation is possible both the REI and RES task.

## 2 RELATED WORK

A number of methods aim for parameter efficient adaption of large (language) models, which adapt a subset or an additional set of the parameters. LoRA (Hu et al., 2022) and its variants (Zhang et al., 2023; Lialin et al., 2023; Liu et al., 2023a; Wu et al., 2024; Sheng et al., 2023; yang Liu et al., 2024) add a trainable residual low rank adaption for each matrix in the network, potentially quantizing it (Dettmers et al., 2024; Xu et al., 2024; Li et al., 2024). In contrast, sparse methods (Ben Zaken et al., 2022; Ansell et al., 2021) only adapt small subsets of the parameters. Adapter based methods (Pfeiffer et al., 2020) train adapter layers and yet another approach is to train a completely separate ladder side networks (Sung et al., 2022; Mercea et al., 2024). As we aim to adapt large multimodal models online, we use LoRA (Hu et al., 2022) for adaptation.

For adapting an MLLM to obtain a desired functionality, such as the ability to adapt to a listener online, different RL methods (Snell et al., 2023; Ziegler et al., 2019; Ramamurthy et al., 2023) can be used. Proximal policy optimization (PPO) (Schulman et al., 2017) is an on-policy actor critic algorithm, which is extended by NLPO (Ramamurthy et al., 2023). It restricts the action space to a nucleus of most likely tokens. In contrast KTO (Ethayarajh et al., 2024) directly optimizes the LLM from binary preferences. On the other hand DPO (Rafailov et al., 2024) requires positive and negative pairs for the same context. All of the methods apart from DPO use a single reward per generation making them suitable for our task, thus, we compare their performance. Similar to our work (Guo et al., 2024; Liu et al., 2024) perform (online) adaption based on model feedback in the context of generic model alignment, while we focus on personalization to individual conversational partners and their misunderstandings.

Personalizing generative language models has been studied for a long time, often viewed in the context of building an efficient conversational partner in dialogue systems (Serban et al., 2015; Song et al., 2019; Zhang et al., 2019). In contrast, (Ma et al., 2023) reviews several theory of mind (TOM) based approaches to personalization, such as (Takmaz et al., 2023) which proposes a plug-and-play TOM based on an explicit simulator, that updates a copy of the model weights on the fly. Similarly, (Raileanu et al., 2018) internally models the behavior of the listener. In contrast, we only update a small amount of parameters using LoRA and do not need to simulate the listeners behavior. (Wang et al., 2024a) adapts the speaker and listener differently, but studies the text-only task, whereas we consider a multi-modal image reference game. We follow an online approach, while (Ma et al., 2021; Zhong et al., 2022) personalizes chatbots by learning from large-scale user dialogue history.

Image identification tasks have been studied in visual dialogue settings in (de Vries et al., 2016; Ni et al., 2021; Alaniz et al., 2021; Das et al., 2016). Our work extends this, by incorporating impairments in the communication. (Corona et al., 2019) has studied conceptual image understanding through a reference game, but we extend their attribute constrained setting to free text generation.

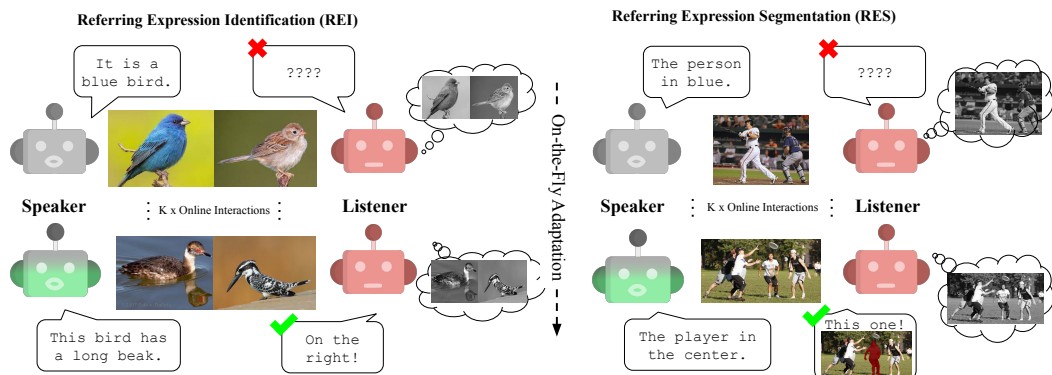

Figure 1: The speaker tries to identify a target object, but its pre-trained policy is not aware of misunderstandings of the listener agents, e.g., color blindness. Through interaction with the listener, the speaker learns on-the-fly to mention the shape instead of color because the listener is color-blind. The left interaction illustrates the REI task, while the right interaction shows the RES task.

## 3 ADAPTING THE SPEAKER ON THE FLY IN REFERRING EXPRESSION TASKS

We present a framework for referring expression communication tasks (Figure 1) where a speaker agent describes images to a listener agent using visual concepts. The "speaker" is a single learner that participates in sequences of $K$ episodes describing an image to a group of "listeners".

**Referring Expression Identification (REI) Task.** In the REI task, each episode involves the speaker $\pi^{(s)}$ and listener $\pi^{(l)}$ being presented with a pair of images $[x_k^t, x_k^c]$. The speaker is assigned one image as the target $x_k^t$ and the other as a confounding image $x_k^c$. The speaker then generates a description $m_k^{(s)}$ as a message to the listener for it to make its guess regarding the target's identity $m_k^{(l)}$, i.e. left or right image. The speaker will observe whether its description led to a correct or incorrect guess via a reward $r_k \in \{+1, -1\}$ communicated for every episode.

**Referring Expression Segmentation (RES) Task.** In the RES task, the speaker $\pi^{(s)}$ and listener $\pi^{(l)}$ are presented with a single image $x_k$ in each episode. The speaker additionally receives the bounding box of a target object $o_k^t$ for which the speaker generates a description $m_k^{(s)}$ with the intention to identify the object in the context of the image. Given the speaker's message, the listener generates a segmentation mask $m_k^{(l)}$ as a guess regarding the target object in the image. The intersection over union (IoU) metric between the predicted and the ground-truth segmentation masks serves as a reward of the episode for the speaker.

Based on this feedback from the reward alone, the speaker's goal is to change its policy $\pi^{(s)*}$, i.e., adapt its image description, to maximize the success rate of the listener agent to solve the referring expression task. To further increase the difficulty of each task, any listener may suffer from a perceptual weaknesses, i.e., color blindness or blurry vision, which is unknown to the speaker.

Since the listener operates as a black box from the perspective of the speaker, pinpointing the source of errors when they exhibit unexpected behavior can be challenging. When the listener makes an incorrect guess, identifying the source of the error becomes difficult, e.g., it could be a lack of comprehension in language, or the visual concepts used to describe the image.

When the listener fails to guess the correct object, the speaker should explore different descriptions to find a policy tailored for the listener. In this work, we examine, whether LLM adaptation methods can successfully find policies that maximize task performance for a diverse set of listener agents solely from the reward signal in this multimodal, i.e. vision and language-based, framework.

### 3.1 ONLINE MLLM ADAPTATION

To perform well on the referring expression tasks, the speaker agent needs to adapt to the listener in an online setting during ongoing interactions. After each episode the speaker can update its

Figure 2: Speaker is asked to describe an object in the context of the REI or RES task. The description is passed to the Listeners which need to decide which image was described. Depending on the correctness of the decision, the Speaker receives a sparse reward and updates its LoRA weights to maximize the reward. For each type of Listener, we have a distinct set of LoRA weights.

weights based on the reward provided by the listener's response. Through these rewards, the speaker increases the likelihood of generating descriptions that are adapted to the capabilities of the listener.

Reinforcement learning from human feedback (RLHF) (Ouyang et al., 2022; Christiano et al., 2017; Stiennon et al., 2020; Ahmadian et al., 2024) is a popular technique to adapt LLMs to human preferences. Typically, a dataset of human preferences is collected, before a RLHF algorithm is applied either offline or through training a reward model to update the parameters of the LLM or MLLM for better human alignment. In this work, we explore how well RLHF algorithms extend to an online setting which is more challenging because the reward data is not carefully annotated and can be noisy, e.g., when the listener misunderstands the description, but still guesses correctly.

**Proximal Policy Optimization (PPO)** (Schulman et al., 2017) is an on-policy actor-critic algorithm that treats language generation as Markov Decision Process (MDP) where at each state $s_t$ in the sequence (current context), the next action $a_t$ is chosen (token), until at the end of the sequence $T$ a reward $r$ is observed. As is typical in RL, the discounted expected reward of the policy is optimized $\mathbb{E}_\pi[\sum_{t=0}^{T} \gamma^t r(s_t, a_t)]$ with $\gamma$ as the discount factor. PPO starts from the initially pre-trained MLLM $\pi_\theta = \pi_0$ and updates the policy using the following loss:

$$\mathcal{L}_{\text{PPO}}(\pi_{\theta_k}, \pi_{\theta_{k-1}}) = \mathbb{E}_{a_t, s_t \sim \pi_{\theta_k}} \left[ \min \left( \phi_{\pi_{\theta_{k-1}}}^{\pi_{\theta_k}} A^{\pi_{\theta_{k-1}}}, \text{clip}(\phi_{\pi_{\theta_{k-1}}}^{\pi_{\theta_k}}, 1 - \epsilon, 1 + \epsilon) A^{\pi_{\theta_{k-1}}}) \right) \right] \quad (1)$$

where $\phi_{\pi_{\theta_{k-1}}}^{\pi_{\theta_k}} = \frac{\pi_{\theta_k}(a_t|s_t)}{\pi_{\theta_{k-1}}(a_t|s_t)}$, $\epsilon$ is a hyperparameter and $A^{\pi_\theta}$ is the advantage function that estimates whether the current action is better than average.

As suggested by Wu et al. (2021), a token-level penalty $\text{KL}(\pi_q||\pi_p) = (\log \pi_p(a_t|s_t) - \log \pi_q(a_t|s_t))$ regularizes the reward function. This avoids large deviations from the pre-trained MLLM, i.e. the initial policy $\pi_0$. The updated reward is computed as:

$$\hat{r}(s_t, a_t) = r(s_t, a_t) - \beta \text{KL}(\pi_\theta||\pi_0) \quad (2)$$

where the KL coefficient $\beta$ is a hyperparameter.

**Natural Language Policy Optimization (NLPO)** (Ramamurthy et al., 2023) extends PPO by restricting the action-space with a reduced number of tokens. This is achieved by freezing a masked policy $\pi_\psi$ every $\mu$ steps and sampling sentences during training from this masked policy. NLPO employs top-$p$ sampling for $\pi_\psi$ which limits the sampled tokens to the smallest subset of tokens with cumulative probability greater than the probability $p$. This additional constraints restricts the sampled sentences to be closer to the masked policy, a snapshot of a previous policy, preventing large deviations and divergence.

**Kahneman-Tversky Optimization (KTO)** (Ethayarajh et al., 2024) takes inspiration from prospect theory and proposes to directly optimize the LLM from binary preferences similar to

DPO (Rafailov et al., 2024), instead of performing RLHF. In contrast to DPO, it does not require paired preference data. The loss function is defined as:

$$L_{\mathrm{KTO}}^{+}(\pi_\theta, \pi_0) = \mathbb{E}_{a_t,s_t \sim \pi_\theta}[\lambda^+(1 - \sigma(\beta(\log \phi_{\pi_0}^{\pi_\theta} - \mathbb{E}_{s' \sim \pi_\theta}[\mathrm{KL}(\pi_\theta \| \pi_0)])))] \quad \text{if } r = +1 \quad (3)$$

$$L_{\mathrm{KTO}}^{-}(\pi_\theta, \pi_0) = \mathbb{E}_{a_t,s_t \sim \pi_\theta}[\lambda^-(1 - \sigma(\beta(\mathbb{E}_{s' \sim \pi_\theta}[\mathrm{KL}(\pi_\theta \| \pi_0)] - \log \phi_{\pi_0}^{\pi_\theta})))] \quad \text{if } r = -1 \quad (4)$$

that depends on whether a generated sentence produced a +1 or -1 reward. $\lambda^{+/-}$ are hyperparameters for the two loss terms respectively. Since we do not have a static dataset, we sample sentences on-policy and shuffle the context, i.e. image input and prompt, within each batch for the KL term.

RL algorithms are known to be unstable (Ouyang et al., 2022; Christiano et al., 2017; Ahmadian et al., 2024) which is why KL terms have been introduced for fine-tuning LLMs. Nonetheless, a potential danger that can arise from this is that the policy of the speaker may diverge and start to generate unusual sentences which exploit the listener agent. These sentences may not describe the images correctly, or deviate from being grammatically correct, but enumerations of words instead. Careful selection of hyperparameters is generally important for success with any of these algorithms.

### 3.2 Efficient Adaptation of the Speaker Agent

Online adaptation of an MLLM does not only require a suitable optimization algorithm, but it should also be feasible in terms of update speed and flexibility as a common use-case may involve a speaker agent interacting with several listeners in parallel. As full-fine-tuning MLLMs is computationally expensive, we adapt these methods by using a parameter-efficient fine tuning method. Given the versatility of LoRA (Hu et al., 2022) for both the visual domain and the text domain, and its simplicity, we employ it in our architecture. We add LoRA adapters on each linear layer in the LLM-module of the network. As a result, the total number of tuneable parameters are orders of magnitude smaller than the total number of parameters in the MLLM. One can initialize one set of LoRA adapters for each listeners and effortlessly swap out LoRA parameters when interacting with multiple listeners.

We employ LLaVA-7B as the speaker model for all experiments because it fits into the memory of a single GPU while training with LoRA adapters. Since the listener runs in inference mode, we also evaluate on LLaVA-13B, Qwen , and PaliGemma to increase listener diversity.

## 4 Experiments

We first introduce our experimental setting, i.e. our datasets, the agents, the training, and evaluation protocol. Then we present the weaknesses and strengths of current MLLMs when dealing with the visual-language referring expression tasks. Finally, we provide extensive experiments into adapting a speaker model to different listeners on four different datasets using three algorithms.

### 4.1 Experimental Setting

**Datasets.** We propose a framework for referring expression tasks on four datasets: CLEVR(Johnson et al., 2017), CUB (Wah et al., 2011), ImageNet(Deng et al., 2009) for REI, and RefCOCO (Kazemzadeh et al., 2014) for RES. CLEVR contains images with objects of varying attributes (size, color, material), requiring fine-grained reasoning to distinguish between different CLEVR scenes. CUB and ImageNet feature natural images with more conversationally relevant concepts. For REI on these datasets, we sample two images, randomly select one as the target, and ask the speaker to describe it in contrast to the other image. We shuffle their order when presenting the images to the listener to avoid trivial solutions, such as "the left image is the target image". Further, we ensure the images come from different classes for CUB and ImageNet. For RES, we employ RefCOCO which extends COCO (Lin et al., 2014) with human-annotated referring expressions and bounding box/segmentation mask annotations. This task requires contrasting a specific detail within an image's context, posing a different challenge from REI. To visually prompt the speaker on the target object, following Shtedritski et al. (2023) we use a red circle as big as the ground truth bounding box.

**Agents.** Our experiments consider pairs of agents: a speaker and a listener. Specifically, we use LLaVA-1.5-7B (Liu et al., 2023b) as the speaker across all adaptation experiments, providing a good balance between its pre-trained capabilities to bootstrap from and a model size that allows us

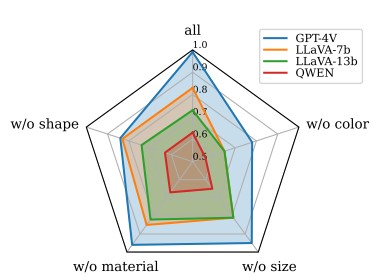

Figure 3: Performance for various agents on ground-truth descriptions with all attributes and with sets of three attributes for CLEVR.

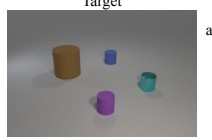

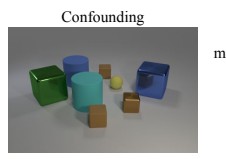

| | |
|---|---|
| All attributes | The image contains a small purple rubber cylinder, a small cyan metal cylinder, a small blue rubber cylinder and a large brown rubber cylinder. |
| w/o color | The image contains a small rubber cylinder and a small metal cylinder. |
| w/o size | The image contains a purple rubber cylinder, a cyan metal cylinder and a brown rubber cylinder. |
| w/o material | The image contains a small purple cylinder, a small cyan cylinder, a small blue cylinder and a large brown cylinder. |
| w/o shape | The image contains a small purple rubber object, a small cyan metal object, a small blue rubber object and a large brown rubber object. |

Figure 4: Example of ground-truth descriptions (right) on CLEVR for the target image (left) with all attributes, and with sets of three attributes.

to fine-tune LoRA adapters on a single A100 40GB GPU. As listener agents, we employ LLaVA-1.5-7B, LLaVA-1.5-13B, Qwen (7B)(Bai et al., 2023) for REI, and PaliGemma (3B) (Beyer et al., 2024) for RES, which is the only open model of reasonable size capable of producing segmentation masks as output. Each listener model has distinct capabilities when it comes to image and language recognition, with Qwen being the weakest one. This diversity in listener agents simulates a population of agents, testing the speaker's ability to adapt its language effectively.

To introduce an additional challenge, we induce perceptual weaknesses in the listener agents: "color blindness" (grayscaled images) and "blurred vision" (Gaussian blur). These weaknesses require the speaker, which receives unaltered images, to adapt its language to account for concepts that are not recognizable by the listener agent.

**Training and evaluation.** We train the speaker (LLaVA-7B) with LoRA adapters on all linear layers of the LLM, keeping the vision module fixed. During online adaptation, we play three episodes before updating the parameters using PPO, NLPO, or KTO algorithms, resulting in a batch size of 3 which maximizes our memory usage. The speaker is trained for 1800 interactions (600 update steps) and evaluated on a held-out test set of 300 episodes per dataset. We use the average success rate as evaluation metric for REI and mean IoU for the RES task. Each experiment combines a specific speaker-listener pair either with or without perceptual weaknesses. We provide additional details about the MLLM prompts in Supp. B.

## 4.2 EVALUATING LISTENERS WITH GROUND-TRUTH DESCRIPTIONS ON CLEVR

CLEVR's detailed scene descriptions allow us to construct a ground-truth (GT) speaker agent for the REI task that produces image descriptions with perfect perception and reasoning abilities. This enables us to evaluate listeners given an ideal speaker. The produced descriptions mention all attributes that appear at least once in the target image, but do not exist in the confounding image. We also ablate the GT speaker by omitting one attribute type, measuring the importance of each attribute for REI. Examples of these image descriptions are shown in Fig. 4.

We evaluate our listener agents alongside GPT-4V, to obtain a reference for a state-of-the-art MLLM, and present the results in Fig. 3. We observe that when all attributes are present, GPT-4V performs best (0.99), followed by LLaVA-7B and LLaVA-13B (0.83 and 0.73), with QWEN being the weakest model (0.63) . Removing size and material attributes has little impact on performance, except for a slight increase in LLaVA-13B and QWEN's scores, indicating that size information is more confusing than helpful for these models, possibly because of perspective. In contrast, omitting shape information significantly affects GPT-4V's performance (from 0.99 to 0.84), while the other listeners are less affected, showing that GPT-4V is more sensitive to shape than other models.

Most notably, removing color information results in significant performance drops across all listeners, highlighting its importance for solving the REI task on CLEVR. These findings demonstrate that different MLLMs prioritize different attributes and have varying capabilities, as shown in Fig. 3. Even GPT-4V struggles to solve the task without color or shape information.

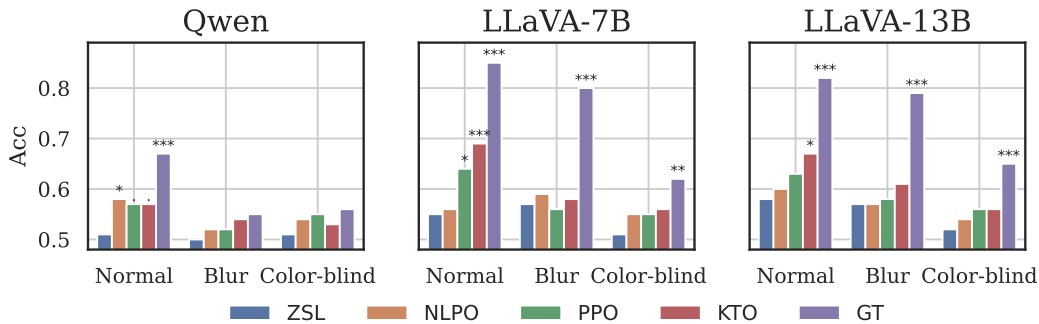

Figure 5: Comparing NLPO, PPO, KTO, GT on CLEVR. ZSL: no training was involved. Normal: no perceptual impairment, Blur: Blurry vision, Color blind: Vision with no color. P-value of statistical significance test w.r.t. ZSL: . $(< 0.1)$, * $(< 0.05)$, ** $(< 0.01)$, *** $(< 0.001)$

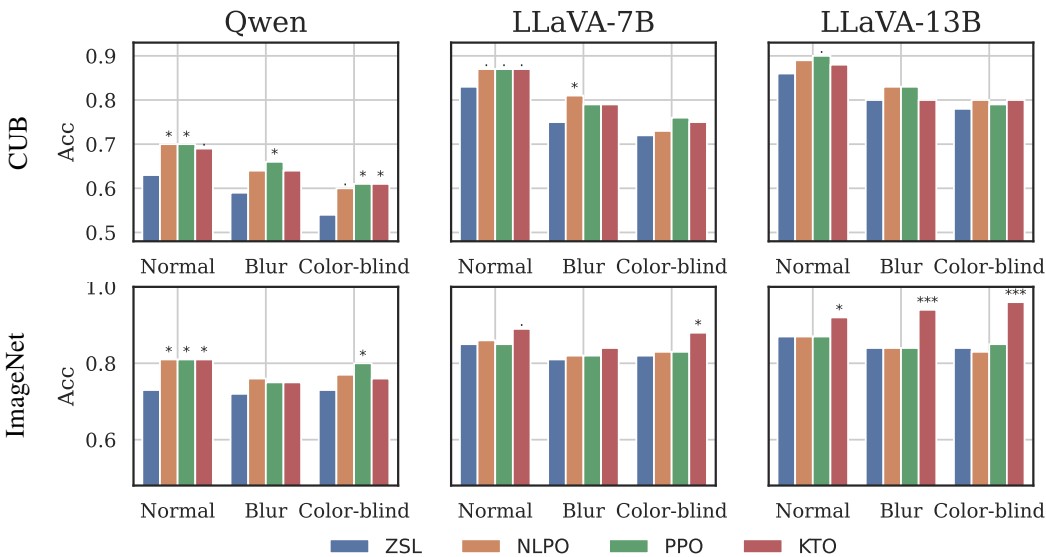

Figure 6: Results on the CUB (Top) and ImageNet (Bottom) datasets (REI task). ZSL means that no training was involved. Perceptual weakness refers to the visual impairment applied to the listener. P-value of statistical significance test w.r.t. ZSL: . $(< 0.1)$, * $(< 0.05)$, ** $(< 0.01)$, *** $(< 0.001)$

### 4.3 COMPARING LISTENERS AND ADAPTATION METHODS ON REI TASK

**REI on CLEVR.** As shown in Fig. 5, when we do not adapt the speaker in the zero-shot learning (ZSL) setting, listener models achieve modest performance. The LLaVA-13B listener achieves the highest performance with an accuracy of 0.58. Introducing color blindness decreases performance for both LLaVA models, while blurred vision has little impact. Qwen performs weakest both with and without perceptual weaknesses, i.e., it struggles to understand the descriptions of LLaVA-7B.

KTO-based adaptation significantly improves performance for LLaVA-7B and LLaVA-13B (peaking at 0.69 and 0.67). Qwen also sees smaller improvements to 0.57. PPO-based adaptation yields smaller gains, while NLPO shows little improvement over zero-shot learning, except when Qwen is the listener. Testing these algorithms with perceptual weaknesses reveals reduced performance increases due to the harder task for the speaker. Blurred vision is generally easier to handle than color blindness, with KTO performing the best overall.

Compared to using GT descriptions for evaluating the listeners (0.67/0.82/0.85), there is a significant gap to the best adaptation results with KTO (0.57/0.67/0.69) even with normal vision. This suggests that the REI task is challenging enough for further research in online adaptation of MLLMs.

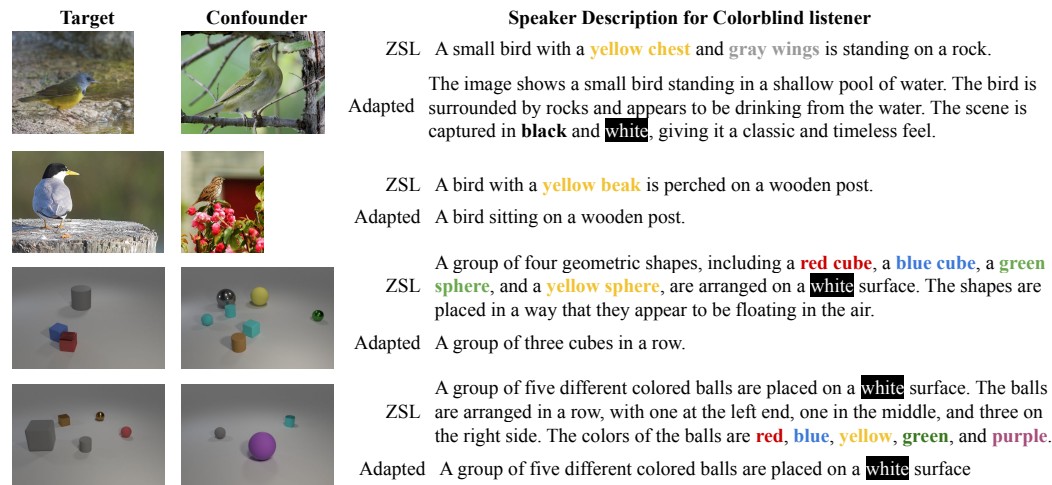

| Target | Confounder | Speaker Description for Colorblind listener |
|---|---|---|

ZSL   A small bird with a **yellow chest** and **gray wings** is standing on a rock.

Adapted   The image shows a small bird standing in a shallow pool of water. The bird is surrounded by rocks and appears to be drinking from the water. The scene is captured in **black** and **white**, giving it a classic and timeless feel.

ZSL   A bird with a **yellow beak** is perched on a wooden post.

Adapted   A bird sitting on a wooden post.

ZSL   A group of four geometric shapes, including a **red cube**, a **blue cube**, a **green sphere**, and a **yellow sphere**, are arranged on a **white** surface. The shapes are placed in a way that they appear to be floating in the air.

Adapted   A group of three cubes in a row.

ZSL   A group of five different colored balls are placed on a **white** surface. The balls are arranged in a row, with one at the left end, one in the middle, and three on the right side. The colors of the balls are **red**, **blue**, **yellow**, **green**, and **purple**.

Adapted   A group of five different colored balls are placed on a **white** surface

Figure 7: Qualitative results on CUB and CLEVR when the speaker interacts with a colorblind listener. We present the descriptions generated by the untrained agents (ZSL) and the descriptions obtained after training (Adapted). After adaption, the speaker avoids color attributes.

**REI on natural images.**  Fig. 6 presents the adaptation results on CUB and ImageNet using natural images. We observe that all listeners perform well in ZSL, with LLaVA-13B achieving an accuracy of 0.86 (CUB) and 0.87 (ImageNet). The MLLMs are likely more familiar with such natural images making it easier for the speaker to pick out differences and the listener to recognize them. However, there is still a large gap to Qwen with 0.63/0.73 for CUB/ImageNet.

In general, adaptation methods provide a boost in performance for all listeners. While KTO-based adaptation excels on ImageNet, all three algorithms perform similarly well on CUB. Perceptual weaknesses have a larger impact on CUB, with removing color having the highest effect on performance. On ImageNet both weaknesses only slightly decrease the performance. This is consistent across listeners and algorithms.

In conclusion, online adaptation is possible for every tested agent and algorithm on the REI task. However, listener capabilities influence improvements, and different algorithms perform better on different datasets and listeners. Overall, KTO seems to work best when considering all experiments. At the same time, none of the existing algorithms are able to find a policy that achieves results close to the of the GT agent leaving room for improvement. Moreover, we find that adaptation on blurred or grayscale images can reach or surpass zero-shot learning performance on normal images, which is a desirable outcome in scenarios where we want to avoid a disadvantage for agents with perceptual weaknesses. This applies to a lesser degree on ImageNet, and was not generally true on CUB, where achieving this target could be an promising direction within the REI task framework.

## 4.4   ADAPTING TO PALIGEMMA ON THE RES TASK

On the referring expression segmentation task, we adapt the LLaVA-7B speaker to PaliGemma as listener on the RefCOCO dataset. In Fig. 9, we report the mean intersection over union (mIoU) for ZSL, PPO, NLPO, and KTO together with probing the PaliGemma listener with the ground truth (GT) referring expressions created by humans that come with the dataset.

We find that the RES task poses a particular challenge to some adaptation algorithms, because neither PPO or NLPO can significantly improve over the zero-shot descriptions in normal, blurred, and grayscaled images. Only KTO manages to obtain an improvement from 0.34 to 0.44 for normal images, from 0.28 to 0.41 in blurry images, and from 0.28 to 0.40 in grayscale images. At the same time, the GT descriptions still outperform the KTO adapted speaker reaching 0.63/0.56/0.61 mIoU in the three settings respectively. Thus, we conclude that there is still room for improvement for online adaptation to reach closer to human performance.

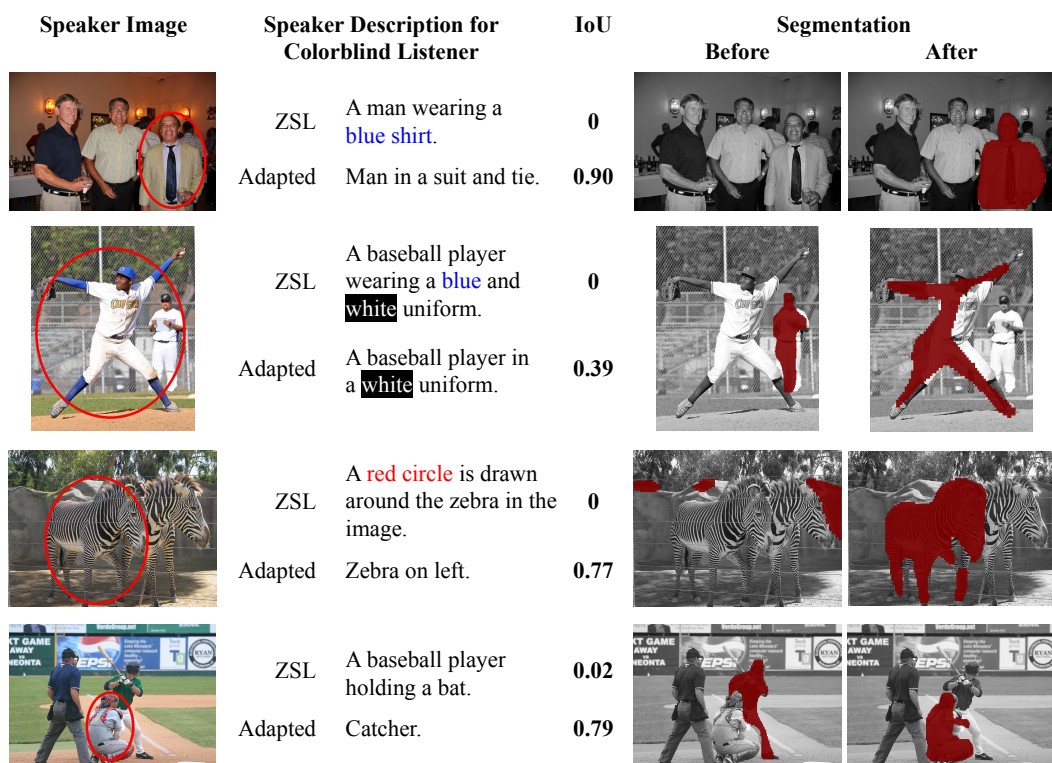

Figure 8: Qualitative results of the RES task on RefCOCO with a LLaVA-7B speaker and coloblind PaliGemma listener.

When inducing perceptual weaknesses on the PaliGemma listener, ZSL performance degrades, but to a lesser degree than for the REI task. This is expected because objects that are contrasted in RES are often easier to identify by their relation in the scene, e.g., where it is located spatially rather than by color or shapes. As a result, PaliGemma can deal with blur and grayscale relatively well in this context. Apart from KTO being the best adaptation algorithm for RES, we also find that KTO can adapt to perceptually weakened listeners to improve over ZSL performance of the normal listener.

Figure 9: mIoU on RefCOCO for RES with LLaVA-7B speaker and PaliGemma listener. P-value w.r.t. ZSL: . $(< 0.1)$, * $(< 0.05)$, ** $(< 0.01)$, *** $(< 0.001)$

### 4.5 QUALITATIVE ANALYSIS ON COLORBLIND LISTENER

In Fig. 7, we show qualitative results on CUB, and CLEVR, by contrasting generated descriptions before and after adaptation on the REI task when interacting with a colorblind listener.

We observe that color attribute is mentioned predominantly before adaptation, and, apart from referring to "black" and "white", completely avoided after adaptation. On CUB for instance, the speaker mentions the "yellow chest" and "yellow beak" to discriminate the birds in the zero-shot setting, and learns to focus the description more on the surrounding scene and action performed by the bird to discriminate the two images after adaptation. On CLEVR, descriptions similarly contains many references to the color attributes in the initial descriptions, but they do not mention colors after adaptation. In contrast, the adapted descriptions focus on the overall count of the objects and are more concise than the original ones. Moreover, zero-shot descriptions sometimes mix objects from both

images, e.g., description in the third row mentions "red cube" and "blue cube" from the left image, and "green sphere" and "yellow sphere" from the right image. After adaptation this behaviour is suppressed and the speaker focuses more on the target image.

In Fig. 8, we show examples of the adaptation on RefCOCO for the RES task, again when the listener is colorblind. The first two rows exemplify how mentioning color can confuse the listener, e.g., in the second row, where the listener segments the incorrect baseball player because it cannot attribute the "blue" uniform to the correct one. After adaptation, not mentioning the colors and focusing on other aspects, such as the "suit and tie" in the first example, allows the listener to more accurately segment the target. Interestingly, there are a few examples where the visual prompting through the red circle (Shtedritski et al., 2023) can cause incorrect descriptions mentioning the circle which is not visible to the listener. However, online adaptation can also correct for this failure case as seen in the third row, where the speaker correctly refers to the "zebra on left".

In conclusion, from these qualitative examples, we observe that the speaker learns to correctly identify the perceptual weakness of the listener, and adapts its description accordingly to be more effective in its communication.

## 5 LIMITATIONS

As it is widely known in the literature (Ouyang et al., 2022; Christiano et al., 2017; Ahmadian et al., 2024), RL algorithms tend to be unstable when the reward signal is noisy, or the actions space immense. During this study, we have observed that there is a divergence effect during online adaptation. Fig. 10 exemplifies this divergence effect on CLEVR dataset for LLaVA-13B which is representative of the observations on other datasets and with other listeners. For all our experiments, we report the performance after 1800 episodes. However, Fig. 10 shows the peak performance is sometimes achieved at different times during training due to the variance in online adaptation. One potential reason for this is the online nature of gathering training samples. The constantly changing policy during training affects the generated data, which in turn influences the future policy and exploration of possible descriptions. With the large actions space of MLLMs, it is challenging to keep these effects in check.

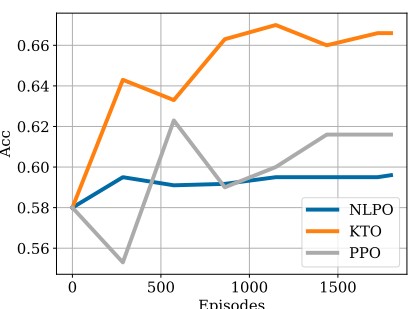

Figure 10: Divergence effect on CLEVR for LLaVA-13B. The performance fluctuates instead of monotonically improving.

## 6 CONCLUSION

In this work, we introduce a framework for two referring expression tasks (REI/RES) involving communicating MLLM agents. On these tasks, we study how MLLM agents can adapt to one another on-the-fly. Our online adaptation setting is significantly more challenging than aligning MLLMs on carefully collected offline datasets, while opening up new applications that require individual personalization. Every communication partner understands language and concepts required to solve the tasks at different levels and we introduce perceptual weaknesses to further control for agent variety. The referring expression tasks pose a challenge to currently available MLLMs, especially for images with fine-grained differences, and when precise segmentation is required. All the adaptation algorithms we have tested could improve task performance on REI with KTO working the best overall and being the only one achieving improvements on the RES task. These results show that, 1) it is possible to improve over the initial pre-trained policy by learning about the listener capabilities, and 2) we can perform this learning in an online setting. However, we also observe that current methods do not monotonically improve during the training process, and cannot find an "optimal" policy, since we have demonstrated that better ones exist with our GT agent experiments. With our task setting, we want to encourage further research on how to make online adaptation of MLLM effective and practically viable to extend to real-world scenarios for MLLM personalization.

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

SUPPLEMENTARY MATERIAL

## A    BROADER IMPACT

In this work we study the capabilities of a speaker to adapt to a listener. We considered MLLMs adapting to other MLLMs, but one could apply these methods also for adapting MLLMs to humans. If such techniques were used to adapt MLLMs to humans, people with malicious intend could purposefully teach the MLLMs to produce harmful or otherwise undesirable content. Online adaptation could effectively overwrite previously learned safety measures of the alignment phase. A possible solution could involve intertwining or following online adaptation with alignment training. Additional research is require to measure both opportunities and risks in this scenario.

In the setting where we adapt an MLLM agent to another MLLM agent, malicious actors could try to exploit systems employing MLLMs by programmatically learning to maximize a desired action of the target MLLM. These "hacks" or "jailbreaks" are a security concern for everyone deploying MLLM, especially if they are deployed adapting to the users. As a result, research on defense mechanisms is just as important as developing more advanced ways to enable personalization.

On the other hand, we believe that allowing MLLMs to adapt to the specific needs of a users can enable new use cases and improve inclusion across diverse population groups. More effective communication towards users with disabilities could lower the barrier of entry and learning curve to bring MLLM technology and their advancement to a broad audience.

## B    MLLM PROMPTING DETAILS

The referring expression identification (REI) task starts with the speaker generating a description for the target image. The prompt given to the speaker is:

```
Write a description for the left/right image, such that it can be
 differentiated from the right/left image, but do not talk about
the right/left image.  Do not name which image you are describing.
```

Subsequently, with the help of the speaker's response, the listener generates a sentence containing its guess. For LLaVA listener agents, we use the query template:

```
Does this sentence:  '{m^{(s)}}' describe the left image or the right
              image?  Do not explain your reasoning.
```

where $\{m^{(s)}\}$ is replaced with the description written by the speaker. On the other hand, Qwen gets the prompt:

```
Which image does the sentence '{m^{(s)}}' describe?  A. Picture 1 B.
                         Picture 2.
```

After receiving the listener's answer, the reward is computed by looking for keywords, i.e. "left, A, 1" and "right, B, 2", and comparing it with the ground truth label.

For the referring expression segmentation (RES) task, the prompt given to the speaker is:

```
        Write a short description for the highlighted object.
```

The PaliGemma listener is then prompted with:

$$\text{segment } '\{m^{(s)}\}'$$

where "segment" is a PaliGemma specific keyword to induce its segmentation capabilites. The model proceeds to output tokens that can be translated to a segmentation mask. We calculate the intersection over union (IoU) between the predicted segmentation mask and the ground truth segmentation mask as a reward for the speaker. Since KTO requires a binary reward, we binarize the IoU values with a threshold of 0.5.

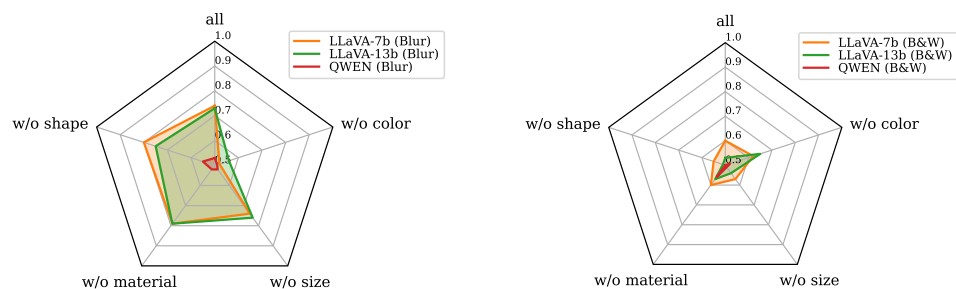

Figure 11: Performance for ground-truth descriptions with blurred vision (left) and colorblindness (right).

Since LLaVA models can only take a single picture as input, we concatenate the images horizontally and add a white bar between them before feeding them to LLaVA-7B and LLaVA-13B. As a result of this step, LLaVA refers to the images as left or right. No such processing is necessary with Qwen, as it can handle multiple images in a single query. Qwen automatically labels them as picture 1 and picture 2.

## C  GROUND-TRUTH DESCRIPTIONS WITH PERCEPTUALLY WEAKENED LISTENERS

We present the evaluation of the GT speaker against listeners with perceptual weakness in Fig. 11. We observed that both blurry and grayscale images cause a significant drop in performance, with the latter having the greatest impact.

When all attributes are mentioned, blurring decreases the scores of LLaVA-7B and Qwen from 0.83 and 0.63 to 0.74 and 0.53. LLaVA-13B maintains its accuracy of 0.73. When the speaker additionally does not mention any color attributes in the description, the accuracy of all listeners drop to near-random performance (i.e., 0.5), with LLaVA-13B performing best at 0.56 accuracy. This result indicates that colors are vital for agents with blurry vision. Removing shapes from the descriptions increases the scores by a small margin in all cases, which suggests this information could be confusing in the presence of blur. Additionally, LLaVA models gain a few percent accuracy when materials are not mentioned in the description. Finally, we would like to highlight that Qwen achieves at most 0.55 score in this setup, which is very close to random guessing.

With grayscale images, LLaVA-7B achieves the lowest score of 0.55 when shape information is lacking in the descriptions, and has the highest accuracy of 0.62 with colors removed. The worst and best cases for LLaVA-13B are again without shape (0.51) and without color (0.65), which have a larger difference compared to the smaller version of LLaVA. Those results show color information starts to confuse the models as it is useless, and mentioning shape is more important in this case. Similar to blurry images, Qwen has a very low performance, with a maximum score of 0.56. These observations support our previous findings that shape and color are the most important attributes for performing well on the REI task with CLEVR images.

### C.1  ADDITIONAL QUALITATIVE RESULTS ON REI

In Fig. 12 we show qualitative results for the REI task on CLEVR, CUB and ImageNet by contrasting generated descriptions before and after adaptation. In CLEVR the original description is much longer and even if the speaker is able to mention all the objects in the image, the associated shapes and color are oftentimes incorrect. On the other hand, after adaptation, the descriptions are much shorter, mentioning a subset but distriminative part of the scene. The adapted policy frequently mentioning shapes ("blocks", "balls") and colors ("yellow and silver") provides additional evidence that these attributes are important and easier to recognize for MLLMs in this context.

The ZSL descriptions generated for CUB images are generic and long, often applying to both images. The speaker tends to confuse the confounding image into the description, for instance, when

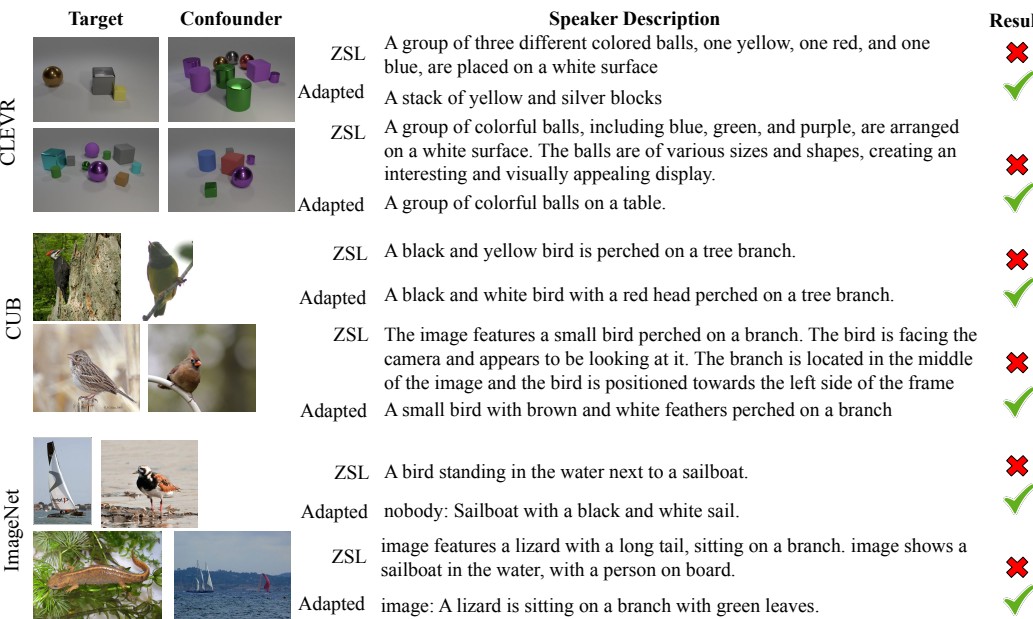

Figure 12: Qualitative results for CLEVR, CUB and ImageNet datasets. We present the descriptions generated by the untrained agents (ZSL) and the descriptions obtained after training (Adapted).

talking about the bird "facing the camera" and the "black and yellow" bird mixing the colors of both birds. In contrast, the trained agent just mentions the essential distinguishable aspects of the target images ("brown and white feathers" and "black and white bird with a red head"). Lastly, on ImageNet, one failure case of the untrained speaker is that it describes both images without clearly identifying the target. After training, it learns to focus on describing the content of the target image by itself. In conclusion, from these qualitative examples, we observe that the model learns to be more concise, focusing on the correct image and primarily mentions the relevant attributes, which more frequently include color and shape.

## D    COMPUTATIONAL RESOURCES

For every experiment, we use 2x A100 40GB GPUs, where one GPU is used for the listener and the other for the speaker. Since the speaker is trained, it requires more computational resources than the listener. It is possible to fit a 13B parameter model into the memory of a single GPU in inference mode for the listener. However, training MLLM only allows models up to 7B parameters on a single GPU, even when using a parameter-efficient fine-tuning method such as LoRA. The training time depends on the lengths of sentences LLaVA generates as the speaker. Longer token sequences take more time to produce as well as to backpropagate through the model. While the length of generations usually diminishes as the speaker adapts to the listener, we also observe the the generated descriptions vary in lengths for the different dataset. Overall, a single experiment of playing 1800 REI episodes and performing 600 update steps (batch size 3) takes around 5-6 hours training time.

## E    HYPERPARAMETERS

For all experiments, we perform a grid search over a subset of hyperparameters and report the results of the best set of hyperparameters. Generally, there was no single set of hyperparameters that performed well across all experiment. The hyperparameters that we considered for grid search are: the learning rate $lr$, the rank $r$ of the LoRA and the $\alpha$ parameters in LoRA. Depending on the algorithms, datasets and models, the $lr$ was searched in the interval [1e-7, 1e-8, 1-9], the $r$ was searched in the interval [32, 64, 128] and the $\alpha$ was searched in the interval [64, 128, 256, 512,

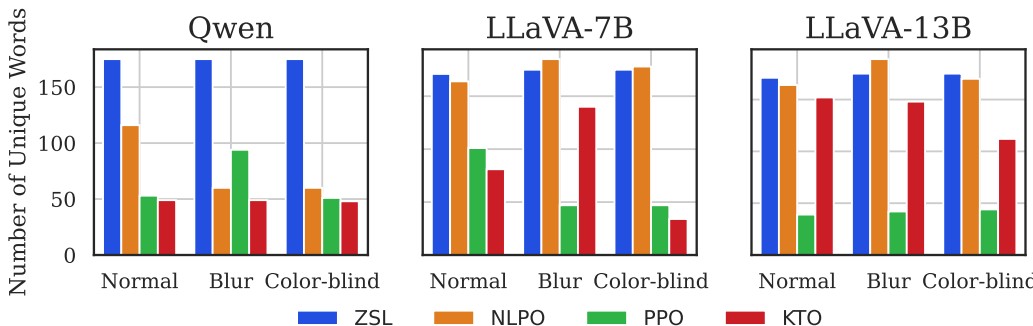

Figure 13: Number of unique words produced by the LLaVA-7B speaker before (ZSL) and after (NLPO, PPO, KTO) adaptation to different listeners.

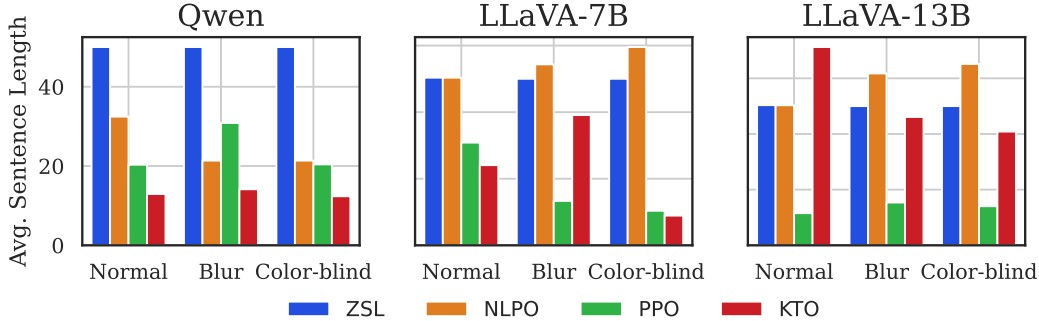

Figure 14: Average sentence length produced by the LLaVA-7B speaker before (ZSL) and after (NLPO, PPO, KTO) adaptation to different listeners.

1024, 2048]. The remaining hyperparameters were kept fixed without performing a grid search. Specifically, for $\beta$ in KTO we used 0.1, and for PPO and NLPO we used 0.2. $\epsilon$ in PPO was set to 1, top-$p$ sampling in NLPO was set to 0.9. $\lambda^-$ and $\lambda^+$ were set to 1.0.

## F   ANALYSIS OF ADAPTED LANGUAGE

We analyze the language of the LLaVA-7B speaker before and after adaptation. Figure 13 shows the number of unique words, i.e., the speaker's vocabulary size when interacting with the different listeners. We find that PPO consistently reduces the number of unique words the speaker uses. However, when interacting with the LLaVA listeners, NLPO does not change the vocabulary size of the speaker. Similarly, KTO also retains the number of unique words when interacting with LLaVA-13B.

Figure 14 shows the average sentence length of the LLaVA-7B speaker. The statistics follow a similar trend to the unique words. Interestingly, NLPO and KTO can even increase the average sentence length, especially when interacting with LLaVA-13B.

In this study, our goal is to adapt to a given listener which can include a change in language characteristics, such as avoiding color words for the color-blind listener. This typically leads to a more concise and effective communication between MLLMs. If language drift should be avoided as much as possible, the hyperparameter for the KL term of each adapatation algorithm can be increased.

## G   QWEN2-VL EXPERIMENTS AND TASK DIFFICULTY

We extend our analysis to include Qwen2-VL-7B Wang et al. (2024b) as a recent open MLLM which is generally stronger than the models evaluated in the main paper. In Table 1, we report the ZSL

| Image Pairing | Normal | B&W | Blur | Occlusion |
|---|---|---|---|---|
| Random | 0.96 | 0.69 | 0.92 | 0.78 |
| Equal #obj. & overlap | 0.95 | 0.56 | 0.89 | 0.68 |
| Min. 8 objects | 0.89 | 0.57 | 0.85 | 0.62 |

Table 1: ZSL performance of Qwen2-VL on the REI task as both speaker and listener on CLEVR for all impairments. Different image pairing strategies alter the difficulty of the task. Normal: no perceptual impairment, Blur: Blurry vision, B&W: Vision with no color, Occlusion: Part of image not visible.

| | LLaVA-7B | | |
|---|---|---|---|
| Qwen2-VL | Normal | Blur | B&W |
| ZSL | 0.71 | 0.66 | 0.54 |
| KTO | 0.72 | 0.66 | 0.56 |
| PPO | 0.74 | 0.66 | 0.56 |

Table 2: Results of the REI task on the CLEVR dataset. Qwen2-VL-7B is the speaker and LLaVA-7B the listener. ZSL means that no training was involved. Normal: no perceptual impairment, Blur: Blurry vision, B&W: Vision with no color.

performance of Qwen2-VL as both speaker and listener on the REI task. The first row (random) is the standard evaluation setting where we randomly sample two images from the CLEVR dataset. We observe that it performs significantly better than any other MLLM reaching close to a perfect score both without impairment and even with the blurry impairment. Additionally, we include experiments on the occlusion impairment as described in Section H. To increase the difficulty of our proposed task, we can alter the sampling of the image pairs. For example, in the second row we only sample images with an equal number of objects and, for every episode, pick one our of 1000 image pairs for which there is the most overlap in identical objects in the scene. This increases difficulty such that the results for the colorblind listener drops from 0.69 to 0.56. Another option is to always sample images with at least 8 objects which is equally challenging for the colorblind listener and also increases difficulty for all other settings, e.g., listener with occlusion impairment drops from 0.78 to 0.62. Overall, while strong MLLMs can often achieve a high zero-shot learning performance on the REI task, we can increase its difficulty by sampling hard image pairs.

In Table 2, we adapt a Qwen2-VL speaker to a LLaVA-7B listener the REI task on CLEVR. We observe that it is generally more challenging to adapt a strong MLLM, such as Qwen2-VL. There are small improvements when adapting Qwen2-VL on a listener without impairment (+3%) or a colorblind listener (+2%), but no improvement on a listener with blurry vision.

## H  Occlusion Impairment

To extend the number of impairments, we explore occlusion as another option. For this impairment, we remove part of both images for the REI task on the listener side. Specifically, we black out the left side of the image up to a given ratio. In Table 3, we report ZSL experiments with Qwen2-VL and LLaVA-7B as both the speaker and listener. We observe that for Qwen2-VL as the speaker, occluding half of the image already reduces performance while for LLaVA-7B this only happens starting from 60% occlusion. As such, occlusion could be used as another type of impairment, for which we leave adaptation experiments to future work.

## I  Qualitative Examples of Failure Cases

Figure 15 shows examples of adaptation on REI tasks for the CUB and CLEVR datasets, where the trained model fails to produce descriptions that help the colorblind listener make correct guesses even after adaptation. Similar to Figure 7, we show descriptions before and after adaptation.
In the zero-shot setting, the captions include color information, which the listener cannot perceive. After adaptation, the models often removes color references, but sometimes fails to make any other adjustment to include supplementary information to make the images distinguishable. For example,

|  | | Occlusion Ratio | | | | |
| Speaker / Listener | 0 | 0.5 | 0.6 | 0.7 | 0.8 | 0.9 |
|---|---|---|---|---|---|---|
| Qwen2-VL / Qwen2-VL | 0.97 | 0.83 | 0.78 | 0.69 | 0.55 | 0.54 |
| Qwen2-VL / LLaVA-7B | 0.70 | 0.58 | 0.55 | 0.51 | 0.49 | 0.52 |
| LLaVA-7B / Qwen2-VL | 0.60 | 0.61 | 0.53 | 0.54 | 0.53 | 0.52 |
| LLaVA-7B / LLaVA-7B | 0.55 | 0.55 | 0.54 | 0.51 | 0.51 | 0.51 |

Table 3: ZSL performance of different speaker-listener pairs on the CLEVR dataset when occluding the left half of each image in the REI task.

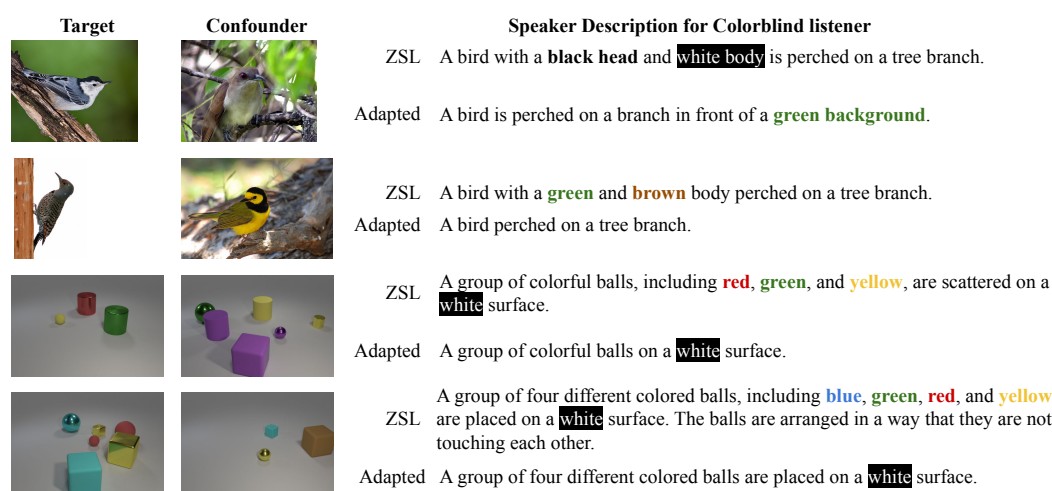

Figure 15: Qualitative results on CUB and CLEVR when the speaker interacts with a colorblind listener and the decision of the listener was wrong after adaptation.

in the first and third rows, both images match the adapted captions, making it hard for the listener to choose the correct one. This suggests that while adaptation helps in some ways by removing color information, the model cannot always introduce other relevant information.

## J    RESULT TABLES

|  | all | w/o shape | w/o material | w/o size | w/o color |
|---|---|---|---|---|---|
| Qwen | 0.63 | 0.63 | 0.67 | 0.65 | 0.56 |
| LLaVA-13b | 0.73 | 0.74 | 0.82 | 0.81 | 0.65 |
| LLaVA-7b | 0.83 | 0.83 | 0.85 | 0.81 | 0.65 |
| GPT-4V | 0.99 | 0.84 | 0.96 | 0.95 | 0.78 |

Table 4: Performance for various agents on ground-truth descriptions with all attributes and with sets of three attributes for CLEVR.

In Tables 4, 5, 6, 7, and 8, we report the results from Figures 3, 5, 6, and 9, respectively.

| LLaVA-7B | LLaVA-7B Normal | Blur | B&W | LLaVA-13B Normal | Blur | B&W | Qwen Normal | Blur | B&W |
|---|---|---|---|---|---|---|---|---|---|
| ZSL | 0.55 | 0.57 | 0.51 | 0.58 | 0.57 | 0.52 | 0.51 | 0.50 | 0.51 |
| NLPO | 0.56 | 0.59 | 0.55 | 0.60 | 0.57 | 0.54 | 0.58 | 0.52 | 0.54 |
| PPO | 0.64 | 0.56 | 0.55 | 0.63 | 0.58 | 0.56 | 0.57 | 0.52 | 0.55 |
| KTO | 0.69 | 0.58 | 0.56 | 0.67 | 0.61 | 0.56 | 0.57 | 0.54 | 0.53 |
| GT | 0.85 | 0.80 | 0.62 | 0.82 | 0.79 | 0.65 | 0.67 | 0.55 | 0.56 |

Table 5: Results of the REI task on the CLEVR dataset. LLaVA-7B is the speaker. ZSL means that no training was involved. Normal: no perceptual impairment, Blur: Blurry vision, B&W: Vision with no color.

| LLaVA-7B | LLaVA-7B Normal | Blur | B&W | LLaVA-13B Normal | Blur | B&W | Qwen Normal | Blur | B&W |
|---|---|---|---|---|---|---|---|---|---|
| ZSL | 0.83 | 0.75 | 0.72 | 0.86 | 0.80 | 0.78 | 0.63 | 0.59 | 0.54 |
| NLPO | 0.87 | 0.81 | 0.73 | 0.89 | 0.83 | 0.80 | 0.70 | 0.64 | 0.60 |
| PPO | 0.87 | 0.79 | 0.76 | 0.90 | 0.83 | 0.79 | 0.70 | 0.66 | 0.61 |
| KTO | 0.87 | 0.79 | 0.75 | 0.88 | 0.80 | 0.80 | 0.69 | 0.64 | 0.61 |

Table 6: Results of the REI task on the CUB dataset. LLaVA-7B is the speaker. ZSL means that no training was involved. Normal: no perceptual impairment, Blur: Blurry vision, B&W: Vision with no color.

| LLaVA-7B | LLaVA-7B Normal | Blur | B&W | LLaVA-13B Normal | Blur | B&W | Qwen Normal | Blur | B&W |
|---|---|---|---|---|---|---|---|---|---|
| ZSL | 0.85 | 0.81 | 0.82 | 0.87 | 0.84 | 0.84 | 0.73 | 0.72 | 0.73 |
| NLPO | 0.86 | 0.82 | 0.83 | 0.87 | 0.84 | 0.83 | 0.81 | 0.76 | 0.77 |
| PPO | 0.85 | 0.82 | 0.83 | 0.87 | 0.84 | 0.85 | 0.81 | 0.75 | 0.80 |
| KTO | 0.89 | 0.84 | 0.88 | 0.92 | 0.94 | 0.96 | 0.81 | 0.75 | 0.76 |

Table 7: Results of the REI task on the ImageNet dataset. LLaVA-7B is the speaker. ZSL means that no training was involved. Normal: no perceptual impairment, Blur: Blurry vision, B&W: Vision with no color.

| LLaVA-7B | PaliGemma Normal | Blur | B&W |
|---|---|---|---|
| GT | 0.63 | 0.56 | 0.61 |
| ZSL | 0.34 | 0.28 | 0.28 |
| NLPO | 0.34 | 0.29 | 0.29 |
| PPO | 0.34 | 0.28 | 0.29 |
| KTO | 0.44 | 0.41 | 0.40 |

Table 8: Results of the RES task on the RefCOCO dataset as IoU with target segmentations. GT refers to providing human annotated referring expressions. For other experiments LLaVA-7B is the speaker. ZSL means that no training was involved. Normal: no perceptual impairment, Blur: Blurry vision, B&W: Vision with no color.

