# OpenReview forum: "Adapting Communicating MLLMs on the Fly in Referring Expression Tasks"
_ICLR.cc/2025/Conference — Submitted to ICLR 2025_

### Official Review · Reviewer_ppz8 · 2024-10-31

**Soundness:** 2
**Presentation:** 2
**Contribution:** 2
**Rating:** 6
**Confidence:** 2

**Summary:**

This paper, "Adapting Communicating MLLMs on the Fly in Referring Expression Tasks," explores whether multimodal large language models (MLLMs) can adapt to their communication partners’ perceptual weaknesses, such as color blindness or blurred vision, in real time. Using referring expression identification (REI) and segmentation (RES) tasks, the paper evaluates how well MLLMs can fine-tune their responses to improve interaction with varying levels of listener comprehension. Through on-the-fly reinforcement learning and using LoRA adapters for efficient fine-tuning, the authors test different MLLMs (LLaVA, Qwen, and PaliGemma) and adaptation algorithms (PPO, KTO, NLPO) on datasets such as CLEVR, CUB, ImageNet, and RefCOCO. The results indicate that online adaptation, especially through KTO, enhances task performance and communication efficacy for MLLMs, revealing the potential for personalized MLLM applications.

**Strengths:**

Originality: The concept of online, real-time adaptation to perceptual weaknesses through reinforcement learning in MLLMs is innovative and provides a step forward for personalized multimodal interactions.
Quality: The methodology is comprehensive, with experiments covering diverse datasets and models, adding to the robustness of the findings.
Clarity: The explanation of the reinforcement learning algorithms (PPO, KTO, NLPO) is well-articulated, as is the application of LoRA for efficient parameter tuning.
Significance: This work addresses a vital aspect of real-time communication adaptation for AI models, potentially making them more inclusive and functional in real-world applications.

**Weaknesses:**

Limited Adaptability: While KTO shows improvement, the adaptation results vary across different tasks and MLLMs, and the paper lacks an exploration of methods to enhance consistency across different perceptual impairments.
Lack of Human Interaction: Although the study uses MLLM-MLLM interactions, the paper could be strengthened by experiments involving human listeners, which would provide a clearer perspective on practical applications.
Evaluation Scope: The paper could further assess performance over a broader range of perceptual weaknesses beyond color blindness and blur, such as partial occlusion or noise.

**Questions:**

Could the authors elaborate on the variance between different adaptation algorithms across datasets, especially why KTO performed better in RES tasks?
Were there any attempts to test the trained models with real human interactions? This could validate the practical applicability of the proposed methods.
How would the proposed method handle more complex perceptual challenges, like occlusion, or scenarios with multiple perceptual weaknesses simultaneously?

---

> ### Author Response · Authors · 2024-11-25
> **Author Response to Reviewer ppz8**
>
> We would like to thank the reviewer for the constructive comments. We are happy that the reviewer finds our study innovative and that it addresses a vital aspect of real-time communication adaptation for AI models. Below, we respond to the concerns and questions raised by the reviewer.
>
> > Limited Adaptability: While KTO shows improvement, the adaptation results vary across different tasks and MLLMs, and the paper lacks an exploration of methods to enhance consistency across different perceptual impairments.
>
> > Could the authors elaborate on the variance between different adaptation algorithms across datasets, especially why KTO performed better in RES tasks?
>
> While the different RL algorithms have been developed to tackle shortcomings of previous methods, their downstream performance on new problems is not always predictable. For instance, NLPO was designed as an extension to PPO specifically for language models but fails to consistently outperform PPO on our tasks. The inductive biases of human-aware losses that motivate KTO seems to better align with our tasks, especially RES. The development of a method that is consistent across tasks and impairments remains an open research question that we would like to tackle as future work.
>
> > Lack of Human Interaction: Although the study uses MLLM-MLLM interactions, the paper could be strengthened by experiments involving human listeners, which would provide a clearer perspective on practical applications.
>
> > Were there any attempts to test the trained models with real human interactions? This could validate the practical applicability of the proposed methods.
>
> Due to the difficulty of designing such a human study which requires care in finding appropriate test subjects with common and diverse disabilities, we did not perform a human study at this time. However, we acknowledge the importance in validating the practical applicability as a future direction.
>
> > Evaluation Scope: The paper could further assess performance over a broader range of perceptual weaknesses beyond color blindness and blur, such as partial occlusion or noise.
>
> > How would the proposed method handle more complex perceptual challenges, like occlusion, or scenarios with multiple perceptual weaknesses simultaneously?
>
> The limited rebuttal time period didn’t allow us to experiment with multiple perceptual weaknesses, however, we include an initial exploration of partial occlusion in Section H of the supplementary. The following table shows the ZSL performance of different speakers and listeners on the REI task and CLEVR dataset. The columns indicate the partial occlusion ratio of the images provided to the listener. Interestingly, even occluding half of the image does not seem to affect the performance when LLaVA-7B is the speaker.
>
> | Speaker / Listener | 0 | 0.5 | 0.6 | 0.7 | 0.8 | 0.9 |
> | --- | --- | --- | --- | --- | --- | --- |
> | Qwen2-VL / Qwen2-VL | 0.97 | 0.83 | 0.78 | 0.69 | 0.55 | 0.54 |
> | Qwen2-VL / LLaVA-7B | 0.70 | 0.58 | 0.55 | 0.51 | 0.49 | 0.52 |
> | LLaVA-7B / Qwen2-VL | 0.60 | 0.61 | 0.53 | 0.54 | 0.53 | 0.52 |
> | LLaVA-7B / LLaVA-7B | 0.55 | 0.55 | 0.54 | 0.51 | 0.51 | 0.51 |

---

> > ### Author Response · Authors · 2024-11-30
> > **Friendly Reminder to Engage in Discussion**
> >
> > Dear Reviewer ppz8,
> >
> > We hope our response has addressed your concerns and we are open to discuss further. If our response is satisfactory, we kindly ask if you would consider raising your score.
> >
> > Thank you again for your time and your valuable feedback.

---

### Official Review · Reviewer_USqB · 2024-10-31

**Soundness:** 2
**Presentation:** 2
**Contribution:** 2
**Rating:** 5
**Confidence:** 3

**Summary:**

This paper introduces a novel framework for referring expression tasks in which two MLLM agents attempt to communicate with each other to communicate information about images in a scene. The dataset/task, based on CLEVR, CUB, ImageNet and RefCOCO is constructed by sampling two images, with one of those images designated as the target image. The "speaker" is then asked to the describe the "target" image relative to the other image, and the "listener" attempts to identify the target image from the two images. This paper evaluates several speaker and listener MLLMs on this task, as well as fine-tunes the speaker MLLM using reinforcement learning, and demonstrates several effects including that adaptation improves listener performance (KTO adaptation improved the best listener accuracy from 0.58 (LLava-13B) to 0.67 on CLEVR and that certain attributes matter more than others (Color and shape attributes were crucial for performance, with GPT-4V’s accuracy dropping from 0.99 to 0.84 on CLEVR when color was omitted).

**Strengths:**

- The paper presents an original approach by exploring real-time adaptive capabilities in MLLMs using RL to dynamically adjust descriptions in communication tasks. This is a relatively novel direction, and is quite an interesting application domain. I find the idea of introducing perceptual weaknesses into the AI to be quite a novel idea - and of great interest - as I think that very few approaches have focused on specializing to perceptual weaknesses. As far as I am aware, there is no other work that studies the idea of having conversational interactions with visually impaired listeners, and looking at how models are capable of handling such situations.
- The paper clearly demonstrates that online RL-based adaptation can improve performance on the scenario tasks.
- The paper's results and claims are quite clear, and easy to understand, and the data presented (fairly well) supports those claims.

**Weaknesses:**

Given the strengths, there are also several weaknesses:
- There is no statistical analysis of the data, so it's quite challenging to tell if there are statistically significant differences between the methods. While the absolute differences are large, the fact that the dataset size is somewhat small (accuracy over 300 episodes), might lead to relatively high variance in the accuracy metrics, and it would be nice to have that variance reported in the paper (especially when significant differences are claimed: L319, L32, L370, L376, L427).
- It's not really clear to me how challenging this task is. Because the images are selected at random, it seems likely that not that much information needs to be communicated between the agents to correctly select the target image/complete the target task. That seems to contrast with the difficulty of the problem for the agents (with the exception of GPT-4v, which seems to perform quite well at the task, achieving almost 100% accuracy). This suggests to me that with some prompt tuning, open models could achieve much higher accuracies as well. The paper would benefit from an improved approach to selecting hard negatives, which might help increase the difficulty of the task.
- It's not clear if the interactions are actually multi-turn (as indicated by Nx in Figure 1), or if the interactions that the agents have are merely single-turn interactions (as seems to be the case in Figure 7 and Figure 8). While it makes sense to have single-turn interactions for simplicity, I think that claiming that MLLMs are "adapting" in the case of single-turn interactions is quite weak. Ideally, the "conversation" should have more than one turn where the speaker must determine the kind of impairment or confusion that the listener has, and then adjust to that, rather than adjust to global speaker impairments over time.
- Several of the effects mentioned in the paper seem to be caused by poor prompting of the speaker MLLMs, rather than actual failures during the task. For example, the effect of visual prompting mentioned on L494, or the non-specific descriptions in Figure 8. It also seems like the descriptions are generally not comparative (Fig 7) - which seems to indicate that the models aren't taking into account multiple images during the prompting process. GPT-4v is rather robust to these prompting issues, and has considerably better performance, so I wonder if that is the underlying cause of many of the effects in this paper.
- The paper only investigates the LLaVA-7B speaker, and does not look at other speaker agents. It would be nice to see if these effects are generalizable to other speaker agents.

**Questions:**

- In figure 8, the ZSL experiments seem to be quite low-quality captions of the image. It seems like the prompting could have quite large impacts on ZSL performance.
- How precisely are images provided to the LLMs (through tiling, or separate image addition)? Models such as Llava are not designed for multi-image reasoning, and so it is important to correctly work around those limitations.
- Does ZSL performance improve when the prompt indicates that the listener may have some kind of impairment (or the impairment is explicitly specified)?
- Does Figure 10 really show a divergence effect? Is this unexpected, or just an artifact of gradient-based optimization? It seems like in general the trend is increasing, as would be expected from RL agents. Further, does Figure 10 plot validation or test accuracy? If it's plotting test accuracy, this would indicate a significant issue in the evaluation methodology, since the model is being tuned on the test set.
- It would be really helpful if Figure 3 was presented as a table instead of a radar chart. Because the axes have no relationship to each other, the shapes are generally misleading, and the chart makes it quite difficult to understand finer grained performance details.
- In general, some more tables would be appreciated, since locating all of the comparative numbers within the paper is quite time consuming. Further, Figures 5,6, and 9 are impossible to read clearly without knowing the base numbers, and might be better as tables.
- It would be interesting to see some failure cases of the model. What is happening when miscommunications occur?
- How do the chosen reinforcement learning algorithms (PPO, KTO, NLPO) compare in terms of training stability? The results in Fig. 10 seem to be from a single run - are the results different across runs?
- Does the use of LoRA impact the adaptation performance compared to fine-tuning all of the parameters?
- Are there model size effects (i.e. using 7B vs. 13B)?

Some additional minor comments:
- The descriptions of RLHF, along with PPO, KTO, and NLPO in Section 3.1 take up a lot of space, and could be moved to the appendix in favor of additional analysis, qualitative results, or tables.
- Is random performance on the task 0.5 accuracy? If so, it would be nice to explicitly clarify that in the paper (since random performance is mentioned on L840). If not, it would be good to know.
- It would be interesting to investigate a wider range of perceptual weaknesses (for example, resolution, partial occlusion, field of view, focal length (blurring at different depths), spatial distortion, inverted colors, etc.).
- The motivation for the specific dataset selection is somewhat unclear, and it would be good to have improved motivation as to why, precisely, these datasets were chosen.
- How expensive (computationally) are these experiments? How long does the average rollout take, and the average experiment?

---

> ### Author Response · Authors · 2024-11-25
> **Author Response to Reviewer USqB (1/3)**
>
> We would like to thank the reviewer for their review of our work, characterizing our work to explore a novel direction that has interesting applications. Below, we respond to the suggestions and concerns mentioned by the reviewer.
>
> > There is no statistical analysis of the data, so it's quite challenging to tell if there are statistically significant differences between the methods. While the absolute differences are large, the fact that the dataset size is somewhat small (accuracy over 300 episodes), might lead to relatively high variance in the accuracy metrics, and it would be nice to have that variance reported in the paper (especially when significant differences are claimed: L319, L32, L370, L376, L427).
>
> We have updated Figures 5, 6, and 9 to indicate which results show a statistical significant difference with respect to the ZSL results by conducting a two-sample statistical hypothesis test. Furthermore, we verified that all the claims mentioned in your comment indeed show a statistical significant difference. While minor improvements are not necessarily significant, we find that the main results we pointed out are significant.
>
> > It's not really clear to me how challenging this task is. Because the images are selected at random, it seems likely that not that much information needs to be communicated between the agents to correctly select the target image/complete the target task. That seems to contrast with the difficulty of the problem for the agents (with the exception of GPT-4v, which seems to perform quite well at the task, achieving almost 100% accuracy). This suggests to me that with some prompt tuning, open models could achieve much higher accuracies as well. The paper would benefit from an improved approach to selecting hard negatives, which might help increase the difficulty of the task.
>
> As mentioned in the global response, we tested Qwen2-VL on the REI task. It performs significantly better than LLaVA which is why we explored increasing the task difficulty on Qwen2-VL. In Section G of the supplementary, we discuss sampling strategies that increase the task difficulty on CLEVR. In the table below we report the ZSL performance of Qwen2-VL as both speaker and listener. While the task performance is quite high in the standard setting of randomly sampling two images (1st row), it decreases as we introduce additional constraints. Sampling images with the same number of objects containing a subset of identical objects (2nd row) as well as sampling images with at least 8 objects (3nd row) increases task difficulty especially for the color blindness and occlusion impairment. We believe our framework allows for enough flexibility to increase the task difficulty if desired.
>
> | Image Pairing | Normal | B&W | Blur | Occlusion |
> | --- | --- | --- | --- | --- |
> | Random	| 0.96 | 0.69 | 0.92 | 0.78 |
> | Equal \#obj. \& overlap   | 0.95 | 0.56 | 0.89 | 0.68 |
> | Min. 8 objects	| 0.89 | 0.57 | 0.85 | 0.62 |
>
> > It's not clear if the interactions are actually multi-turn (as indicated by Nx in Figure 1), or if the interactions that the agents have are merely single-turn interactions (as seems to be the case in Figure 7 and Figure 8). While it makes sense to have single-turn interactions for simplicity, I think that claiming that MLLMs are "adapting" in the case of single-turn interactions is quite weak. Ideally, the "conversation" should have more than one turn where the speaker must determine the kind of impairment or confusion that the listener has, and then adjust to that, rather than adjust to global speaker impairments over time.
>
> We focus on single-turn interactions in this study, with Nx referring to the number of episodes (updated to Kx for consistency). While we acknowledge the importance of multi-turn adaptation for complex conversations as a future goal, we note that training and evaluating such free-form interactions without prior knowledge of impairments is challenging. Our evaluation demonstrates that even single-turn interactions pose difficulties for current models and RL methods. Our primary objective is to permanently adapt the speaker's policy to listener impairments, ensuring effective communication from the start of every conversation. In contrast, in-context adaptation through multi-turn interactions would require re-discovering the impairment each time, which we argue is a less desirable goal.

---

> ### Author Response · Authors · 2024-11-25
> **Author Response to Reviewer USqB (2/3)**
>
> > Several of the effects mentioned in the paper seem to be caused by poor prompting of the speaker MLLMs, rather than actual failures during the task. For example, the effect of visual prompting mentioned on L494, or the non-specific descriptions in Figure 8. It also seems like the descriptions are generally not comparative (Fig 7) - which seems to indicate that the models aren't taking into account multiple images during the prompting process. GPT-4v is rather robust to these prompting issues, and has considerably better performance, so I wonder if that is the underlying cause of many of the effects in this paper.
>
> > In figure 8, the ZSL experiments seem to be quite low-quality captions of the image. It seems like the prompting could have quite large impacts on ZSL performance.
>
> We explored various prompts throughout this study to ensure that the observed effects are not solely due to prompt-related issues. However, it's worth noting that LLaVA-7B was not trained on multiple images as input, such that we concatenate images into a single image. Additionally, these models were not natively trained on visual prompting (i.e. providing a visual cue instead of a text prompt), which explains the mentioning of the red circle as part of the image. As such our tasks are challenging for current open MLLMs. Nevertheless, our results show that through adaptation, the speaker model learns to adapt its descriptions to avoid mentioning aspects that the listener does not perceive.
>
> > The paper only investigates the LLaVA-7B speaker, and does not look at other speaker agents. It would be nice to see if these effects are generalizable to other speaker agents.
>
> We refer the reviewer to the global response where we address this comment.
>
> > How precisely are images provided to the LLMs (through tiling, or separate image addition)? Models such as Llava are not designed for multi-image reasoning, and so it is important to correctly work around those limitations.
>
> To accommodate LLaVA's limitations in handling multiple images, we found that concatenating the images horizontally with a white bar separating them yields the best results (as mentioned in the supplementary material, Section B). Therefore, we pass a single image containing both original images to LLaVA.
>
> > Does ZSL performance improve when the prompt indicates that the listener may have some kind of impairment (or the impairment is explicitly specified)?
>
> We tested specifically adding the explicit impairment to the prompt. Specifically, we tried the following prompts, e.g., for the color blind listener:
> - “Write a description for a colorblind person for the left/right image, such that it can be differentiated from the right/left image.”
> - ”I am colorblind. Write a description for the left/right image, such that it can be differentiated from the right/left image.”
> - “Write a description for the left/right image, such that it can be differentiated from the right/left image by a colorblind person.”
>
> None of the prompts could improve the baseline performance with the descriptions being largely unchanged and still mentioning the color attributes of the objects. Hence, we conclude that online adaptation with parameter optimization is more effective.
>
> > Does Figure 10 really show a divergence effect? Is this unexpected, or just an artifact of gradient-based optimization? It seems like in general the trend is increasing, as would be expected from RL agents. Further, does Figure 10 plot validation or test accuracy? If it's plotting test accuracy, this would indicate a significant issue in the evaluation methodology, since the model is being tuned on the test set.
>
> The plot in Figure 10 indeed represents test accuracy. However, we'd like to clarify that this doesn't imply an issue with tuning on the test set, as we always report results after 1800 episodes, even though performance might peak earlier and then decline. The purpose of this plot is to highlight that there exists a higher-performing model that could be obtained if training were stopped at the optimal point, potentially due to divergence. We emphasize that this plot was created after training and did not influence any tuning of the model or selection of the checkpoint.
>
> > It would be really helpful if Figure 3 was presented as a table instead of a radar chart. Because the axes have no relationship to each other, the shapes are generally misleading, and the chart makes it quite difficult to understand finer grained performance details.
>
> > In general, some more tables would be appreciated, since locating all of the comparative numbers within the paper is quite time consuming. Further, Figures 5,6, and 9 are impossible to read clearly without knowing the base numbers, and might be better as tables.
>
> We thank the reviewer for their suggestions. We will improve the readability and have provided the result values of all experiments in tables in Section J of the supplementary.

---

> ### Author Response · Authors · 2024-11-25
> **Author Response to Reviewer USqB (3/3)**
>
> > It would be interesting to see some failure cases of the model. What is happening when miscommunications occur?
>
> We show qualitative results for failure cases after adaptation in Figure 15 and provide a discussion in Section I of the supplementary. In summary, we find that, although the speaker learns to remove color information for the colorblind listener, it most often fails when it removes color information without providing new complementary information about the target image. As a result the descriptions after adaptation sometimes match both images as discriminative color attributes were removed.
>
> > How do the chosen reinforcement learning algorithms (PPO, KTO, NLPO) compare in terms of training stability? The results in Fig. 10 seem to be from a single run - are the results different across runs?
>
> We find that the observation in Fig. 10 is representative for all learning algorithms across runs. Performance generally improves, but can fluctuate significantly during the run.
>
> > Does the use of LoRA impact the adaptation performance compared to fine-tuning all of the parameters?
>
> > Are there model size effects (i.e. using 7B vs. 13B)?
>
> Due to computational constraints (A100 GPU with 40 GB), we are unable to perform full finetuning or adapt a 13B model. However, previous work has shown that LoRA is very effective for adaptation tasks across different model sizes (Hu et. al., 2022; Dettmers et al., 2023; Ghosh et al., 2024).
>
> > Is random performance on the task 0.5 accuracy? If so, it would be nice to explicitly clarify that in the paper (since random performance is mentioned on L840). If not, it would be good to know.
>
> Yes, random performance is 0.5 for REI. We clarified it in L840.
>
> > It would be interesting to investigate a wider range of perceptual weaknesses (for example, resolution, partial occlusion, field of view, focal length (blurring at different depths), spatial distortion, inverted colors, etc.).
>
> While the rebuttal time period didn’t allow us to experiment with an extensive set of additional perceptual weaknesses, we include an initial exploration of partial occlusion in Section H of the supplementary. The following table shows the ZSL performance of different speakers and listeners on the REI task and CLEVR dataset. The columns indicate the partial occlusion ratio of the images provided to the listener. Interestingly, even occluding half of the image does not seem to affect the performance when LLaVA-7B is the speaker.
>
> | Speaker / Listener | 0 | 0.5 | 0.6 | 0.7 | 0.8 | 0.9 |
> | --- | --- | --- | --- | --- | --- | --- |
> | Qwen2-VL / Qwen2-VL | 0.97 | 0.83 | 0.78 | 0.69 | 0.55 | 0.54 |
> | Qwen2-VL / LLaVA-7B | 0.70 | 0.58 | 0.55 | 0.51 | 0.49 | 0.52 |
> | LLaVA-7B / Qwen2-VL | 0.60 | 0.61 | 0.53 | 0.54 | 0.53 | 0.52 |
> | LLaVA-7B / LLaVA-7B | 0.55 | 0.55 | 0.54 | 0.51 | 0.51 | 0.51 |
>
> > The motivation for the specific dataset selection is somewhat unclear, and it would be good to have improved motivation as to why, precisely, these datasets were chosen.
>
> The datasets were chosen to cover a range of difficulties. CLEVR was chosen for its fine-grained reasoning in a controlled setting (MLLMs are forced to talk about object properties). CUB and ImageNet are natural image benchmarks representing a more realistic environment while CUB being fine-grained and ImageNet coarse. For RES, RefCOCO is a popular benchmark such that it was a natural choice.
>
> > How expensive (computationally) are these experiments? How long does the average rollout take, and the average experiment?
>
> Computational details are mentioned in Section D of the supplementary. Playing 1800 REI episodes and performing 600 update steps (batch size 3) takes around 5-6 hours on 2x A100 40GB GPUs.
>
> References:
> Edward J. Hu, Yelong Shen, Phillip Wallis, Zeyuan Allen-Zhu, Yuanzhi Li, Shean Wang, Lu Wang,
> and Weizhu Chen. LoRA: Low-Rank Adaptation of Large Language Models. ICML 2021
> Tim Dettmers, Artidoro Pagnoni, Ari Holtzman, and Luke Zettlemoyer. QLoRA: Efficient
> Finetuning of Quantized LLMs. NeurIPS 2023
> Sreyan Ghosh, Chandra Kiran Reddy Evuru, Sonal Kumar, Ramaneswaran S, Deepali Aneja, Zeyu
> Jin, Ramani Duraiswami, and Dinesh Manocha. A Closer Look at the Limitations of Instruction
> Tuning. ICML 2024

---

> > ### Comment · Reviewer_USqB · 2024-11-26
> >
> > I appreciate the comments and clarifications from the authors. While the paper is significantly improved from the first version, and many of my concerns have been addressed, I still hesitate to change my score given the following concerns:
> >
> > - **Significant Effects in the data:** I'd like to thank the authors for including significance testing across the paper. Unfortunately, adding such numbers raises some concerns that bear additional analysis: for example, in Figure 7, across most of the experiments in Imagenet, and several experiments on CUB, there appears to be little evidence that the adaptation process is helping (p < 0.05, with the exception of KTO on LLava-13B). Interestingly, the results on CUB/CLEVR do not always agree with those on ImageNet, and this warrants additional exploration (or at least a justification for why that is the case). For those experiments that do indicate improvement, there's little overall analysis of why such improvement exists. What is it about KTO that lends itself to this task, compared to PPO or NLPO (Figure 9 confirms that there's something unique here, but it's not clear from the writing if there's any intuition for this result)? Given the new statistical significance results, I believe that the discussion section could be notably improved and expanded upon, and such an expansion would be challenging to do thoroughly within the span of a single revision cycle.
> > - **Incomplete Experiment Set:** While I do appreciate the additional experiments provided in the rebuttal, the other reviews, as well as the limited set of experiments the authors ran for the rebuttal, have made it clear to me that the paper would benefit from a more thorough treatment across the axes of speaker/listener (Such as the inclusion of the speaker listener table in the above response).
> > - **Inclusion of informal results:** While I do appreciate that the authors have run several variants of the prompts ("None of the prompts could improve the baseline performance", "We explored various prompts throughout this study to ensure that the observed effects are not solely due to prompt-related issues."), or several variants of experiments (i.e. "We find that the observation in Fig. 10 is representative for all learning algorithms across runs."), these results should be reported and demonstrated in the paper. I think that the paper would benefit from some additional time to run such multiple variants, and adapt the figures as such.
> >
> > Some things that would make this paper much better:
> > - **Multi-turn adaptation:** While I agree with the authors that this is out of scope for the current project, it feels like in most cases, the natural way of solving this task is through multiple turn few-shot adaptation, as opposed to through fine-tuning individual models for specific perceptual deficiencies. Exploring how such an approach compares could easily make this paper much more exciting.
> > - **Expanded perceptual set:** It seems somewhat limiting to only explore blurring or colorblindness. Further expanding the perceptual set would help demonstrate the applicability of this approach more broadly, and help confirm the utility of the proposed methods.
> >
> > Some other questions that could be resolved in a revision:
> > - Why is LLava-13B so much worse on ground truth descriptions than LLaVA-7b on CLEVER?

---

> > > ### Author Response · Authors · 2024-11-28
> > >
> > > We thank the reviewer for their constructive feedback that helps us improve our paper.
> > >
> > > > Significant Effects in the data
> > >
> > > We agree that there is some variation across datasets which is likely attributed to the characteristics of the datasets, each posing a different challenge, fine-grained reasoning for CLEVR, natural images for CUB (fine-grained) and ImageNet (coarse-grained). When there is an improvement on a dataset, it is difficult to pinpoint the reason as the downstream performance of these RL algorithms on new problems is not always predictable. For instance, NLPO was designed as an extension to PPO specifically for language models but fails to consistently outperform PPO on our tasks. It is possible that the inductive biases of human-aware losses that motivate KTO better align with our tasks, especially RES. We believe it is valuable to show that our proposed task is challenging which could also inspire research on developing a method that is consistent across tasks and impairments in an online adaptation setting.
> > >
> > > > Incomplete Experiment; Inclusion of informal results; Some things that would make this paper much better
> > >
> > > We appreciate the constructive suggestions. In the final version, we will incorporate the informal results into the paper and make the Qwen2-VL and Occlusion experiments more complete.
> > >
> > > > Why is LLava-13B so much worse on ground truth descriptions than LLaVA-7b on CLEVER?
> > >
> > > Unfortunately, we couldn't find a good explanation for this behavior and understanding large models to such a detail is still an open research problem. However, we believe the GT descriptions are an outlier and in the other settings, LLaVA-7B and LLaVA-13B are on par or LLaVA-13B is better.

---

### Official Review · Reviewer_jKcf · 2024-11-01

**Soundness:** 2
**Presentation:** 2
**Contribution:** 3
**Rating:** 5
**Confidence:** 5

**Summary:**

In this paper, the authors study the efficient online adaptation of agents implemented as Multimodal Language Models (Vision language models more specifically). Particularly, the authors study this online adaptation with both a reference identification task (REI) and a reference segmentation task (RES). In REI, a listener needs to select the correct target image between 2 images using a description generated by a speaker. For the RES task, the speaker needs to generate a description for an object in an image and the listener has to derive a segmentation mask for it.

In this paper, the authors study how the well-known RLHF algorithms can be adapted to the online setting which is more challenging because they typically refer to single-turn data rather than dialogues with noisier rewards. For their evaluation, they test different SOTA VLMs on images derived from relatively standard benchmarks such as COCO, ImageNet, etc.

The results highlights that adaptation seems to have a negative effect on the quality of the descriptions which diverge to very unnatural ones which do not include any object attributes compared to the Zero-shot variants that are much more descriptive.

**Strengths:**

1. Studying continuous adaptation of Vision and Language Models is definitely an interesting topic that should be explored more by the community

2. The authors test different training regimes using techniques such as KTO, PPO and NLPO. The evaluation uses different models such as Llava-7B and Llava-13B which makes the experiments very reproducible by the community

**Weaknesses:**

1. The authors consider referential games with only two images which incredibly reduces the ambiguity of the task. Additionally, they do not compare with existing literature from multi-agent language evolution (e.g., [1])

2. It's not clear to me to what extent the benchmarks that the authors have used are completely unseen by the models. For instance, it's very likely that RefCOCO is part of the Llava fine-tuning considering that they use MSCOCO images. Authors should pay more attention to the problem of data contamination which I believe was ignored by the authors.

3. The models used in the evaluation are not up to date considering that there are many strong variants such as Llava-1.5, QwenVL-2, Llama-3.2 and Molmo. I would suggest the authors provide additional results with these baselines to make the results much stronger.

4. The authors should clarify the way the different models are adapted. Do they always adapt the speaker or only the listener? This is an important research question that I think is not clearly highlighted by their evaluation.

5. Their models are clearly affected by language drift during the adaptation procedure. I believe the authors should focus on a more detailed analysis of the language developed by the models and how it changes over the different games. This should also be compared to utterance length and vocabulary size to verify whether models are simply maximising success rate and forgetting their language abilities.

## References

[1]: Lazaridou, A., & Baroni, M. (2020). Emergent multi-agent communication in the deep learning era. arXiv preprint arXiv:2006.02419.

**Questions:**

1. Examples in Figure 8 are particularly bad in terms of the quality of the description. How do you explain this? Have you thought about mitigating this language drift problem?

2. Why did you use PaliGemma only for the RES task?

3. It's not clear to me the example in Figure 2. The example seems very unfortunate because it's hard to discriminate two birds when the image is black and white.

---

> ### Author Response · Authors · 2024-11-25
> **Author Response to Reviewer jKcf (1/2)**
>
> We would like to thank the reviewer for the insightful assessment of our work and the helpful comments, indicating that we study an underexplored topic. In the following, we address the concerns and questions raised by the reviewer.
>
> > The authors consider referential games with only two images which incredibly reduces the ambiguity of the task. Additionally, they do not compare with existing literature from multi-agent language evolution (e.g., [1])
>
> We appreciate the reviewer bringing up [1], which actually provides examples similar to our referential tasks (Fig. 3, Fig. 4). This shows that we follow established literature in this area, but in a more complex setting: free-form language and MLLMs. These additions make our scenario more challenging than previous literature, even with only two images. Currently, open MLLMs still have rather limited capabilities when it comes to multi-image comprehension, but it is an interesting research direction to further increase the difficulty as MLLMs improve. We will add this literature to the related works discussion.
>
> > It's not clear to me to what extent the benchmarks that the authors have used are completely unseen by the models. For instance, it's very likely that RefCOCO is part of the Llava fine-tuning considering that they use MSCOCO images. Authors should pay more attention to the problem of data contamination which I believe was ignored by the authors.
>
> While our models may have seen RefCOCO training images during pre-training, this doesn't invalidate our study. Our goal is to adapt the speaker to describe images for listeners with disabilities, not to simply reproduce descriptions for “perfect” listeners as done in pre-training. This task requires out-of-distribution learning, as we need to adapt to the dataset with new labels (responses from listeners with disabilities), making prior knowledge gathered during pre-training less relevant. On top of this, the validation/test splits of these datasets are not part of the pre-training, nor our adaptation, and, thus, still commonly serve as benchmarks for MLLMs, e.g., PaliGemma was evaluated on the RefCOCO benchmark.
>
> > The models used in the evaluation are not up to date considering that there are many strong variants such as Llava-1.5, QwenVL-2, Llama-3.2 and Molmo. I would suggest the authors provide additional results with these baselines to make the results much stronger.
>
> We refer the reviewer to the global response where we address this comment.
>
> > The authors should clarify the way the different models are adapted. Do they always adapt the speaker or only the listener? This is an important research question that I think is not clearly highlighted by their evaluation.
>
> To clarify, we consistently adapt the speaker to the listener, and never adapt the listener to the speaker. We discuss this in Section 4.1 ("We train the speaker...") and Section 3.2 ("Efficient adaptation of the speaker agent"). The idea behind this choice is that the active communication partner (speaker) adapts its language to improve the understanding of the listener and its (fixed) impairments.
>
> > Their models are clearly affected by language drift during the adaptation procedure. I believe the authors should focus on a more detailed analysis of the language developed by the models and how it changes over the different games. This should also be compared to utterance length and vocabulary size to verify whether models are simply maximising success rate and forgetting their language abilities.
>
> We refer the reviewer to the global response where we address this comment.

---

> > ### Comment · Reviewer_ppz8 · 2024-11-27
> > **feedback**
> >
> > Thanks i think most my concerns are answered!

---

> > > ### Author Response · Authors · 2024-11-30
> > >
> > > We are glad that our response has addressed your concerns.
> > > Accordingly, we would appreciate if you would consider raising your score.
> > >
> > > Thank you again for your time and your valuable feedback.

---

> ### Author Response · Authors · 2024-11-25
> **Author Response to Reviewer jKcf (2/2)**
>
> > Examples in Figure 8 are particularly bad in terms of the quality of the description. How do you explain this? Have you thought about mitigating this language drift problem?
>
> We disagree that the examples in Fig. 8 have bad quality. In fact, they cater towards the prompts PaliGemma was trained on, which include mentioning single objects without forming full sentences. Due to RL optimization, the speaker will find a policy that suits the listener. If the listener prefers (was pre-trained) on full sentences only, the adapted speaker should reflect this fact. If one desires to mitigate this effect even for listeners such as PaliGemma, one can increase the weighting factor of the KL term in the RL algorithms which prevents language drift. This was not our primary goal in this study.
>
> > Why did you use PaliGemma only for the RES task?
>
> PaliGemma performed poorly when given multiple images compared to the other MLLMs we tested on the REI task. This could be related to the relatively low parameter count of PaliGemma (3B) compared to the others (7B, 13B). On the other hand, due to its segmentation capabilities (which other models do not have), it suits the RES task well.
>
> > It's not clear to me the example in Figure 2. The example seems very unfortunate because it's hard to discriminate two birds when the image is black and white.
>
> Our intention is that, when we introduce perceptual impairments, the task becomes hard and the most natural descriptions might not work anymore for a given listener. As a result our learning pipeline adapts the speaker to avoid using colors in descriptions for colorblind listeners and instead mentions other attributes. As such, we believe Figure 2 faithfully represents the difficulties presented in this task.

---

### Official Review · Reviewer_enKE · 2024-11-03

**Soundness:** 2
**Presentation:** 3
**Contribution:** 2
**Rating:** 5
**Confidence:** 3

**Summary:**

This paper studies an interesting problem setup: how to adapt MLLMs to perceptual misunderstandings. The paper introduces a novel framework of having a speaker MLLM and a listener MLLM, where the speaker MLLM need to learn adaptation to the listener MLLM on the fly so that the listener MLLM can come up with the correct answer. The paper proposes two settings where the MLLM can have misunderstanding: color blindness and blurry images. The paper tested on 3 RL algorithms to do online adaptation: PPO, NLPO, KTO, and found that KTO attains the best performance. The paper also provides qualitative results of the response difference between the adapted MLLM and the original MLLM.

**Strengths:**

1. The paper proposes an interesting setting of the communication between MLLMs. The paper proposes to have one speaker and one listener, where the speaker need to address the best way to communicate with the listener so that the listener can arrive at the correct answer. This setting is novel.
2. The paper proposes on the fly adaptation based on the listener's feedback. The real time adaptation is interesting and useful.
3. The paper conducts thorough experiments on three RL algorithms and demonstrates the effectiveness of KTO. The experiments provides a thorough comparison between different RL algorithms.
4. The writing is generally clear and easy to follow.

**Weaknesses:**

1. The paper lacks comparison with more baseline methods. Some simple and training-free personalization methods in MLLMs might directly solve this problem better. Eg, adding the experiment of few-shot/one-shot learning would be a useful comparison with the online adaptation method that the paper proposes. RAG methods with a memory augmented module. Or some implict/explicit prompt engineering techniques.
2. The qualitative analysis is not thorough enough. Eg, in Figure 7, the author noted that the adapter-generated description is better because it has less color attributes. However, this is still a surface level analysis as there are many differences between the two descriptions generated, such as length. It would be better to conduct a deeper analysis of the comparison between the adapter generated, such as the response length.
3. The paper covers the scope "color blind" and "blur" as the two attributes of the listener. It is not clear to me how these two attributes are chosen and how they align with real-world misunderstanding between MLLMs.

**Questions:**

1. The ZSL baseline seems insufficient. What about comparison with few-shot learning? eg. Given the trajectory of the past prediction results as context, would the speaker learn to better describe the object?
2. What are the costs of introducing RL learning? Would be useful to add an analysis.
3. It would be interesting to test the speaker's final understanding of the , eg, would the speaker be able to identify that the listener is color-blind in the end?

---

> ### Author Response · Authors · 2024-11-25
> **Author Response to Reviewer enKE**
>
> We would like to thank the reviewer for the constructive feedback, highlighting our contributions in proposing a novel setting for adapting MLLMs on the fly which the reviewer finds interesting and useful. In the following, we respond to the questions in the following.
>
> > The paper lacks comparison with more baseline methods. Some simple and training-free personalization methods in MLLMs might directly solve this problem better. Eg, adding the experiment of few-shot/one-shot learning would be a useful comparison with the online adaptation method that the paper proposes. RAG methods with a memory augmented module. Or some implict/explicit prompt engineering techniques.
>
> > The ZSL baseline seems insufficient. What about comparison with few-shot learning? eg. Given the trajectory of the past prediction results as context, would the speaker learn to better describe the object?
>
> We appreciate the suggestion to compare our approach with simpler baseline methods, specifically in-context learning (few-shot/one-shot learning). However, we argue that this approach may not be feasible for this task due to computational limitations.
>
> For example, consider LLaVA 1.5, which encodes every image into 512 image tokens and has a total context length of 4K tokens. Multiple adaptation interactions in the context would quickly fill up the available token space. Specifically, with only 8 conversations, the image tokens alone would consume all of the context, leaving no room for text tokens.
>
> Furthermore, LLaVA (similar to most current open models) does not support multiple images in a conversation, making it challenging to faithfully provide previous games in-context. Even as we move to more capable models with longer context lengths, the number of image tokens also grows; for instance, LLaVA 1.6 uses 2880 image tokens for a single image while still being trained on only a 4K context length with the underlying LLM supporting a theoretical maximum of 32K tokens.
>
> Regarding one-shot learning, we acknowledge its potential feasibility but highlight the difficulty in choosing a single shot that maximizes information about the disability. A single successful/unsuccessful interaction may not necessarily reveal valuable insights.
>
> Finally, regarding implicit/explicit prompting techniques, it is unclear what form such prompts should take for this task. As the goal of our approach is to enable the speaker to discover and adapt to the listener's needs without prior knowledge, we cannot include information about the listener's disability in the prompt.
>
> > The qualitative analysis is not thorough enough. Eg, in Figure 7, the author noted that the adapter-generated description is better because it has less color attributes. However, this is still a surface level analysis as there are many differences between the two descriptions generated, such as length. It would be better to conduct a deeper analysis of the comparison between the adapter generated, such as the response length.
>
> We refer the reviewer to the global response where we address this comment.
>
> > The paper covers the scope "color blind" and "blur" as the two attributes of the listener. It is not clear to me how these two attributes are chosen and how they align with real-world misunderstanding between MLLMs.
>
> We chose color blindness and blur (myopia) as examples of common human disabilities and because it was previously studied by Corona et al. (2019), but our approach can be applied to other simulated disabilities that affect communication. The key point is that we're tackling the problem of adaptation in a speaker-listener setting, where the listener can be another MLLM or even a human. Our simulation using an MLLM as the listener is just one possible representation, and our framework is meant to generalize beyond this specific setup.
>
> > What are the costs of introducing RL learning? Would be useful to add an analysis.
>
> We provide details on the computational cost in Section D of the supplementary material, where a single experiment involving 1800 REI episodes and 600 update steps (batch size 3) takes around 5-6 hours training time using 2x A100 40GB GPUs, with one GPU dedicated to the listener and the other to the speaker.
>
> > It would be interesting to test the speaker's final understanding of the , eg, would the speaker be able to identify that the listener is color-blind in the end?
>
> Since the speaker implicitly learns to adapt to the listeners impairment, it cannot directly articulate an explainable summary of how its policy has changed in natural language. We believe that explaining the learned policy change concisely is an interesting research question for future work.

---

> > ### Author Response · Authors · 2024-11-30
> > **Friendly Reminder to Engage in Discussion**
> >
> > Dear Reviewer enKE,
> >
> > We hope our response has addressed your concerns and we are open to discuss further. If our response is satisfactory, we kindly ask if you would consider raising your score.
> >
> > Thank you again for your time and your valuable feedback.

---

> > > ### Comment · Reviewer_enKE · 2024-12-03
> > >
> > > Thank you for taking the time to address my questions and for providing additional clarification. While your responses have helped clarify certain aspects of the work, I still believe that the experimental validation and analysis require further depth. For this reason, I will maintain my current score of weak rejection.

---

### Author Response · Authors · 2024-11-25
**Global Author Response**

We thank the reviewers for their insightful comments. As summarized by reviewers enKE, jKcf, USqB, and ppz8, our paper presents a novel framework for adapting Multimodal Language Models (MLLMs) to perceptual misunderstandings in referring expression tasks. Specifically, we introduce a speaker MLLM and a listener MLLM that learn to adapt to each other's strengths and weaknesses on the fly, using reinforcement learning algorithms such as PPO, NLPO, and KTO. Our evaluation on various datasets, including CLEVR, CUB, ImageNet, and RefCOCO, shows that online adaptation improves task performance, particularly with the KTO algorithm. The results also highlight the importance of certain attributes, such as color and shape, in referring expression tasks. To the best of our knowledge, this is the first systematic study on adapting MLLMs to perceptual weaknesses in online communication scenarios.

All reviewers (enKE, jKcf, USqB, ppz8) agree that our submission presents a novel and interesting approach to adapting Multimodal Language Models (MLLMs) to perceptual weaknesses in online communication scenarios. Reviewer enKE highlights the originality of our setting, which proposes a speaker and listener MLLM architecture with on-the-fly adaptation based on listener feedback. Our thorough experiments using three RL algorithms (KTO, PPO, NLPO) on open models are commended by reviewers enKE, jKcf, and ppz8 for their reproducibility and comprehensiveness. Reviewer USqB notes that our work is a novel direction in the field, with few approaches focusing on specializing to perceptual weaknesses. The clarity of our writing and presentation are also praised by all reviewers, making it easy to understand our claims and results. As reviewer ppz8 states, our concept of online, real-time adaptation through reinforcement learning is innovative and provides a step forward for personalized multimodal interactions. With this rebuttal, we would like to further demonstrate the significance and impact of our work, addressing the remaining comments and suggestions from the reviewers.

We address common remarks by the reviewers in the following.

---

> ### Author Response · Authors · 2024-11-25
> **Global Author Response**
>
> ### Language Analysis
>
> #### Reviewer enKE:
> > The qualitative analysis is not thorough enough. Eg, in Figure 7, the author noted that the adapter-generated description is better because it has less color attributes. However, this is still a surface level analysis as there are many differences between the two descriptions generated, such as length. It would be better to conduct a deeper analysis of the comparison between the adapter generated, such as the response length.
>
> #### Reviewer jKcf:
> > Their models are clearly affected by language drift during the adaptation procedure. I believe the authors should focus on a more detailed analysis of the language developed by the models and how it changes over the different games. This should also be compared to utterance length and vocabulary size to verify whether models are simply maximising success rate and forgetting their language abilities.
>
> We added a language analysis before and after adaptation in Section F of the supplementary. Specifically, we inspect the speaker’s number of unique words (i.e. vocabulary) in Figure 13 and average sentence length in Figure 14. In general, both statistics vary depending on the RL method and the listener agent. While for PPO, both statistics generally drop, the metrics remain mostly the same for NLPO and KTO on LLaVA listeners, especially 13B. Notably, NLPO and KTO show an increased sentence length for the LLaVA-13B listener indicating that these methods can find policies where longer descriptions are beneficial. We want to point out that a changing language after adaptation is not necessarily a downside as it can also enhance conciseness or effectiveness in communication by adapting to a specific listener. If it is desirable to more strictly stay close to the initial LLM policy, all methods include a hyperparameter for the KL term that mitigates language drift which could be tweaked accordingly. In this study, we specifically want to allow for a changing language, e.g., avoiding color words for the color-blind listener, so we adopt the standard hyperparameter settings proposed by the RL methods.
>
>
> ### Additional results with Qwen2-VL
> #### Reviewer jKcf:
> > The models used in the evaluation are not up to date considering that there are many strong variants such as Llava-1.5, QwenVL-2, Llama-3.2 and Molmo. I would suggest the authors provide additional results with these baselines to make the results much stronger.
> #### Reviewer USqB:
> > The paper only investigates the LLaVA-7B speaker, and does not look at other speaker agents. It would be nice to see if these effects are generalizable to other speaker agents.
>
> The LLaVA variant we used throughout our study is LLaVA-1.5 which we clarified in Section 4.1. Nonetheless, we have conducted additional experiments with the recently released Qwen2-VL-7B to include a stronger baseline. We discuss the results in Section G of the supplementary. The following table shows the adaptation experiments on the REI task and CLEVR dataset where Qwen2-VL is the speaker and LLaVA-7B the listener. We find that adaptation is more difficult for Qwen2-VL, yielding overall smaller improvements. We believe these initial results could encourage further investigation into online adaptation with our tasks. Unfortunately, we weren’t able to run additional experiments due to the time constraints of the rebuttal.
>
> | Qwen2-VL / LLaVA-7B | Normal | Blur | B&W |
> | --- | --- | --- | --- |
> | ZSL | 0.71 | 0.66 | 0.54 |
> | KTO | 0.72 | 0.66 | 0.56 |
> | PPO | 0.74 | 0.66 | 0.56 |

---

### Meta-Review · Area_Chair_vRDr · 2024-12-23

**Metareview:**

This paper studies whether Vision Language Models (VLM) can adapt to communicate effectively in presence of perceptual weaknesses (color blindness, etc). The paper studies this in the context of two referring expressions task (identification -- REI), and segmentation (RES).
In REI, given a description given by the speaker, a listener has to predict the correct target image between a pair images. In RES task, the speaker generates a description for an object in an image and the listener predicts a segmentation mask for it. The paper shows that it is possible to adapt online using RL.

Reviewers feel that the experimental setup can be made more complex (e.g., more than two images, multi-turn dialog, expanding perceptual set). Reviewers raised concerns about using random pair of images which makes the task easier since the model might not need to communicate a lot of nuanced information to discriminate between the two pair of images. Reviewers believe that a future version of the paper will benefit from a more thorough analysis of prompts, and discussion of variability in improvements across benchmark.

**Additional Comments On Reviewer Discussion:**

- Experiments with training-free personalisation methods like by using few-shot examples . The authors argue that open-source VLMs (LLaVA 1.6 cannot handle multiple images effectively given the limited context length. Additionally, one-shot learning might be ineffective given the difficulty of choosing of one example for each task. The authors also mentioned that since no prior knowledge of listener needs, adapting in an online fashion might not be feasible.

- Experiments with more recent open-weights model: The authors tried experiments with Qwen2-VL-7B. In initial experiments during rebuttal, this model achieves high zero-shot performance on the REI task when image pairs are random. When pairing with LLAVA as listener, the authors found that adaptation is more difficult with this model, and that additional time will be needed to investigate further.

- Surface level analysis: Reviewer enKE was concerned that the qualitative analysis is not thorough, and that to gain insights into where the improvements are coming from, a deeper analysis of successful and failure cases will be useful. The authors improved the submission by providing a few failure cases in the supplmentary along with some quantitative analysis.

---

### Decision · Program_Chairs · 2025-01-22

Reject